# Single-cell transcriptome reveals insights into the development and function of the zebrafish ovary

Yulong Liu[1], Michelle E Kossack[1†], Matthew E McFaul[1†], Lana N Christensen[1], Stefan Siebert[1], Sydney R Wyatt[1], Caramai N Kamei[2], Samuel Horst[1], Nayeli Arroyo[1], Iain A Drummond[2], Celina E Juliano[1], Bruce W Draper[1*]

[1]Department of Molecular and Cellular Biology, University of California, Davis, Davis, United States; [2]Mount Desert Island Biological Laboratory, Bar Harbor, United States

**Abstract:** Zebrafish are an established research organism that has made many contributions to our understanding of vertebrate tissue and organ development, yet there are still significant gaps in our understanding of the genes that regulate gonad development, sex, and reproduction. Unlike the development of many organs, such as the brain and heart that form during the first few days of development, zebrafish gonads do not begin to form until the larval stage (≥5 days post-fertilization). Thus, forward genetic screens have identified very few genes required for gonad development. In addition, bulk RNA-sequencing studies that identify genes expressed in the gonads do not have the resolution necessary to define minor cell populations that may play significant roles in the development and function of these organs. To overcome these limitations, we have used single-cell RNA sequencing to determine the transcriptomes of cells isolated from juvenile zebrafish ovaries. This resulted in the profiles of 10,658 germ cells and 14,431 somatic cells. Our germ cell data represents all developmental stages from germline stem cells to early meiotic oocytes. Our somatic cell data represents all known somatic cell types, including follicle cells, theca cells, and ovarian stromal cells. Further analysis revealed an unexpected number of cell subpopulations within these broadly defined cell types. To further define their functional significance, we determined the location of these cell subpopulations within the ovary. Finally, we used gene knockout experiments to determine the roles of *foxl2l* and *wnt9b* for oocyte development and sex determination and/or differentiation, respectively. Our results reveal novel insights into zebrafish ovarian development and function, and the transcriptome profiles will provide a valuable resource for future studies.

**\*For correspondence:**
bwdraper@ucdavis.edu

†These authors contributed equally to this work

**Competing interest:** The authors declare that no competing interests exist.

## Editor's evaluation

This single-cell transcriptomic analysis of young adult zebrafish ovaries provides important new data to understand gene expression patterns in numerous ovarian cell types that lead to insights into how ovary development works, and most of the principles will likely apply across vertebrates. The work will interest researchers who study gonad development, sex determination, differences (or 'disorders') in sex development, and impacts of the environment (including toxic pollutants) on gonad development and function.

## Introduction

Over the last several decades, the zebrafish has emerged as a model to study vertebrate gonad development and function. Zebrafish ovaries and testes contain homologs of most cell types present in mammalian gonads (e.g., Sertoli and Leydig cells in testes, and follicle and theca cells in ovaries). As

in mammals, the zebrafish ovary and testis form from a gonad primordium that is initially bipotential (reviewed in *Siegfried and Draper, 2020*). The bipotential stage in zebrafish lasts until approximately 15–20 days post-fertilization (dpf) when sex is determined. Though the mechanism by which sex is determined in domesticated zebrafish is not known, zebrafish have orthologs of most genes that drive sex differentiation in mammals. In several cases, mutational analysis has revealed these genes play conserved roles in sex determination and differentiation. Examples include the male-promoting gene *dmrt1* and the female-promoting genes *wnt4*, *foxl2a*, and *foxl2b* (*Kossack et al., 2019*; *Webster et al., 2017*; *Yang et al., 2017*). Though the embryonic origin of the bipotential gonad has yet to be determined in zebrafish, orthologs of genes that are expressed in, and required for, the development of the bipotential gonad in mammals, such as *Gata4* and *Wt1*, are also expressed in the bipotential gonad in zebrafish (*Leerberg et al., 2017*). Together, these data argue that gonad development in vertebrates is regulated by a largely conserved genetic program.

One major difference between zebrafish and mammalian reproduction is that the mammalian ovary produced a finite number of oocytes only during embryogenesis; by contrast, in many teleost (bony fish), such as the zebrafish, adult females can produce new oocytes throughout their lifetime due to the presence of self-renewing germline stem cells (GSCs; *Beer and Draper, 2013*; *Cao et al., 2019b*). Thus, the zebrafish adult ovary contains germ cells at all stages of development, from premeiotic germ cells to mature eggs, while all mammalian oocytes arrest in prophase of meiosis I (also called dictyate arrest) around the time of birth (*Faddy et al., 1987*; *Selman et al., 1993*). The ability to continuously produce new germ cells throughout their lifetime makes female zebrafish an excellent research organism to study female GSCs and the somatic cells that regulate their development.

The unipotent GSCs can divide both to replenish their numbers while also producing cells that differentiate into sex-specific gametes, either eggs or sperm. In organisms where GSCs have been identified, they localize to a special microenvironment called the GSC niche. This niche is crucial for maintaining most GSCs in an undifferentiated state while also allowing some to differentiate. GSCs and their niche have been previously identified in a limited number of organisms, such as *Drosophila*, *Caenorhabditis elegans,* and male mice (*Xie and Spradling, 2000*). Identified GSC niches are maintained by specialized somatic cells that produce short-range signals that regulate GSC proliferation and differentiation. For example, in *Drosophila*, somatic cap cells in the ovary maintain GSCs by expressing specific ligands, such as *decapentaplegic* (*dpp,* mammalian *Bmp*), to prevent GSC differentiation (*Song et al., 2004*; *Xie and Spradling, 2000*), while in the mouse testis, Sertoli cells in the seminiferous tubules express glial-derived neurotrophic factor (GDNF), which is required for GSC self-renewal (*Meng et al., 2000*).

GSCs have been identified in the zebrafish ovary, but little is known about the niche and mechanism by which GSCs are maintained. In the adult ovary, early germ cells localize to a discrete region on the surface of the ovary, called the germinal zone, which has been proposed to be the GSC niche (*Beer and Draper, 2013*). The germinal zone contains mitotically dividing GSCs, oocyte progenitor cells, and early meiotic oocytes (*Beer and Draper, 2013*; *Draper et al., 2007*). However, neither the GSC-intrinsic gene expression landscape nor the extrinsic GSC niche cells that likely regulate early GSC differentiation in zebrafish have been characterized.

The best-characterized somatic cell type in the zebrafish ovary are the follicle cells. Teleost follicle cells are homologous to mammalian granulosa cells and thus are the major oocyte support cell. Follicle and granulosa cells produce signals that regulate oocyte development and maturation, and also produce nutrients that are transported into the oocyte cytoplasm via gap junctions (*Su et al., 2009*). Unlike granulosa cells, which form a multi-cell-layered complex surrounding the mammalian oocyte, follicle cells in fish form only a single-cell layer around the oocytes. In mammals, granulosa cells are the major source of ovarian estradiol (E2) production (*Ryan, 1979*), and this is likely to be the case in zebrafish. In addition to follicle cells, E2 production also requires steroidogenic theca cells that produce androstenedione, the precursor that follicle cells use for E2 synthesis (*Young and McNeilly, 2010*). Theca cells are found around and between oocytes but are otherwise not well characterized in zebrafish. The final major cell type in the ovary are the stromal cells, which are composed of all other somatic cell types present, including connective tissue, blood vessels, and immune cells. The fibroblast-like interstitial stromal cells play a largely structural role, but the roles of the other stromal cell types in ovarian development and function are not well characterized in any vertebrates. Although these general cell populations have been observed in the zebrafish ovary, there are significant gaps

in knowledge regarding the identification of the full diversity of cell types present in the ovary or the functional roles they play.

In this study, we characterize the complex cell environment in the zebrafish ovary using single-cell RNA sequencing. This data has allowed us to identify all known stages of germ cells, from GSCs to pre-follicle stage oocytes (stage IA; *Selman et al., 1993*), as well as the somatic cell populations, including follicle, theca, and stromal cells. Using subcluster analysis, we have defined subpopulations within these broad cell-type classifications and validated these subpopulations by determining where they reside in the ovary using hybridization chain reaction RNA fluorescent in situ hybridization (HCR RNA-FISH). Further, we demonstrated the accuracy of this dataset to identify developmentally relevant genes using gene knockout analysis. Our data provide strong support that orthologs of genes involved in mammalian ovary development and function play parallel roles in zebrafish, further supporting zebrafish as a relevant research animal to understand human ovarian development and diseases. This reference dataset is available in a processed and interactively browsable form through the Broad Institute Single Cell Portal (available here) and will serve as a resource to greatly enhance future studies of ovarian development and function in unprecedented detail.

## Results and discussion
### Single-cell RNA sequencing identifies all cell types present in the zebrafish juvenile ovary

To identify all cell types and states in the zebrafish ovary, we collected single-cell transcriptomes from ~25K cells isolated from 40-day post-fertilization (dpf) ovaries. We chose 40 dpf ovaries for three major reasons. First, by 30 dpf sex determination is complete and the initial bipotential gonad has committed to differentiating as either an ovary or testis (*Kossack and Draper, 2019*). Second, by 40 dpf all major ovarian somatic cell types are likely present (*Rodríguez-Marí et al., 2005*). Finally, as we were particularly interested in understanding how the developmental progression of GSCs is regulated, 40 dpf ovaries have a higher proportion of early-stage germ cells relative to adult ovaries (*Draper, 2012*).

We prepared three single-cell libraries from dissociated Tg(*piwil1:egfp*)$^{uc02}$ transgenic ovaries, where eGFP is expressed in all germ cells (*Figure 1—figure supplement 1*; *Leu and Draper, 2010*). Two of these libraries were prepared using a dissociation method that favored isolation of single somatic cells. The third library was prepared from cells isolated using a milder dissociation method that favored germ cell survival, followed by fluorescence-activated cell sorting (FACS) to purify eGFP+ germ cells (*Figure 1—figure supplement 1*; see 'Materials and methods' for details). We used the 10X Genomics platform for library preparation and sequencing. Following quality control, data cleaning, and the removal of red blood cells, doublets, and ambient RNA, we recovered a total of 25,089 single-cell transcriptomes that comprise 10,658 germ cells and 14,431 somatic cells (*Figure 1—figure supplement 2*; see 'Materials and methods'). We obtained germ cells with an average of 2510 genes/cell, 11,294 transcripts/cell and somatic cells with an average of 854 genes/cell, 4812 transcripts/cell.

Cell clustering analysis grouped cells into eight distinct populations using uniform manifold approximation and projection (UMAP; *Figure 1*, *Figure 1—figure supplement 3A*, and *Supplementary file 1*). We assigned these clusters provisional cell-type identities using the expression of known cell type-specific genes as follows: germ cells – *deadbox helicase 4* (*ddx4*; formerly *vasa*; *Yoon et al., 1997*); follicle cells – *gonadal soma derived factor* (*gsdf*; *Gautier et al., 2011*); theca cells – *cytochrome P450 cholesterol side-chain cleaving enzyme* (*cyp11a2*; *Parajes et al., 2013*); vasculature – *Fli-1 proto-oncogene* (*fli1a*; *Brown et al., 2000*); neutrophils – *myeloid-specific peroxidase* (*mpx*; *Lieschke et al., 2001*); macrophages – *macrophage-expressed gene 1* (*mpeg1.1*; *Zakrzewska et al., 2010*); NK cells – *NK-lysin tandem duplicate 2* (*nkl.2*; *Carmona et al., 2017*; *Tang et al., 2017*; *Figure 1*, *Figure 1—figure supplement 3B*). Except for blood vessels and blood cells, the remaining stromal cells are the most ill-defined cell population in the teleost gonad, and no genes have previously been identified that specifically mark this population. Ovarian stromal cells are generally defined as the cells that are not components of the ovarian follicle (i.e., germ and associated follicle and theca cells) and play structural roles (*Kinnear et al., 2020*). We determined that expression of the gene encoding *collagen, type I, alpha 1a* (*col1a1a*; *Fisher et al., 2003*), is highly enriched in the remaining unidentified population of cells in the UMAP graph. Collagen is well known for its structural role in tissues and therefore

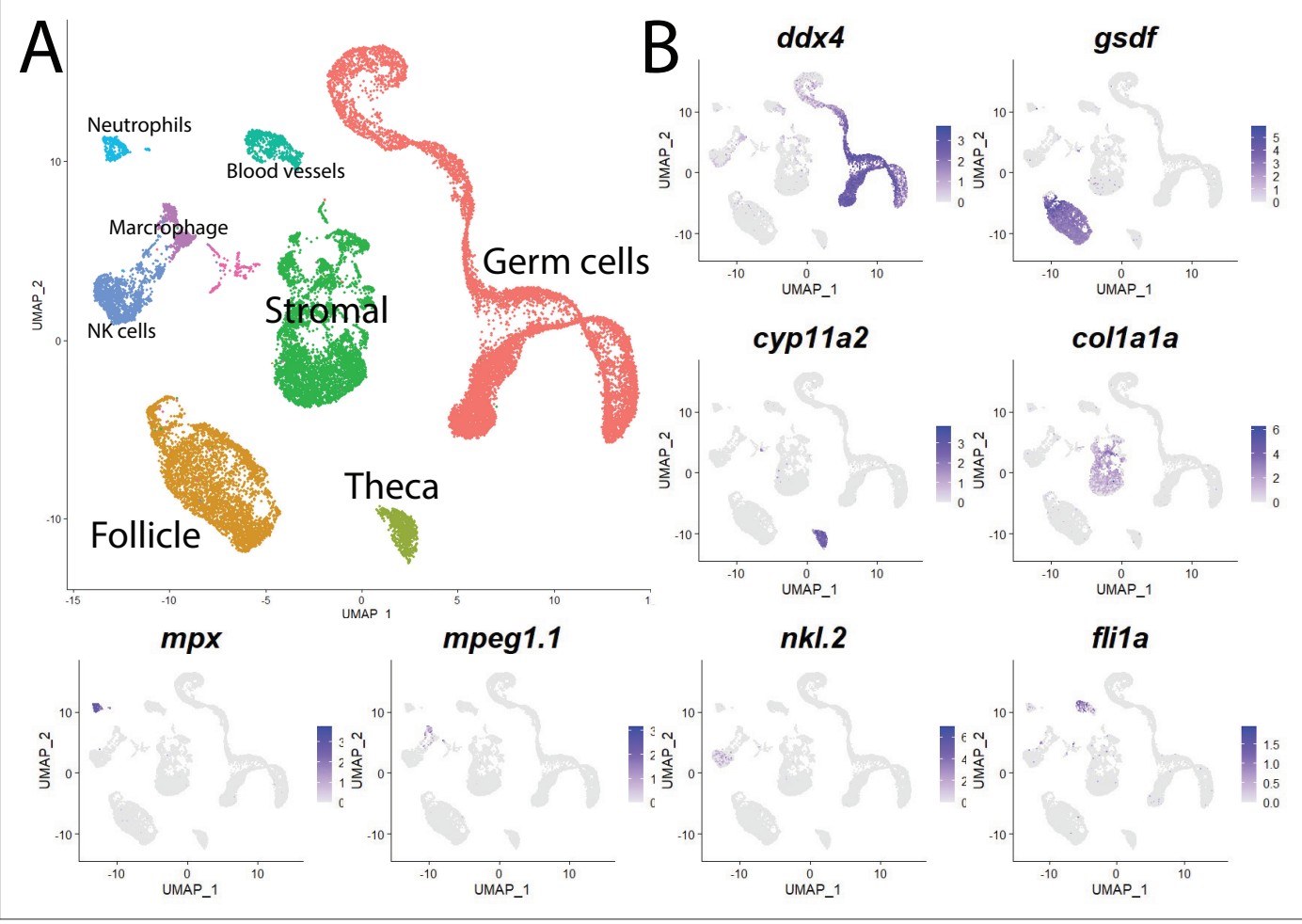

**Figure 1.** Single-cell RNA sequencing of 25,089 single cells isolated from 40-day-old zebrafish ovaries. (**A**) Single-cell uniform manifold approximation and projection (UMAP) plot of the 40-day-old zebrafish ovary. Cells are color-coded by computationally determined cell clusters. (**B**) Gene expression plots of known cell-specific marker genes identify the major cell type (labeled in **A**) that each cluster corresponds to. Cells expressing the indicated gene are colored purple, and the relative intensity indicates relative expression levels (intensity scale for each plot is on the right).

The online version of this article includes the following figure supplement(s) for figure 1:

**Figure supplement 1.** Workflow for single-cell RNA sequencing library preparation, data cleaning, analysis, and validation.

**Figure supplement 2.** Major cell-type statistics.

**Figure supplement 3.** Differential expression heatmap and top markers of major cell types.

reasoned that these cells are ovarian stromal cells (*Figure 1B*, *Figure 1—figure supplement 3B*). Thus, we were able to identify all previously observed and expected cell types in our dataset. A list of genes with enriched expression in each cluster can be found in *Supplementary file 1*.

Our initial analysis revealed genes that are differentially expressed between major cell types but did not distinguish possible subpopulations. For example, *ddx4* and *gsdf* were correctly identified as genes expressed specifically in germ cells and follicle cells, respectively; however, specific genes expressed in discrete developmental stages of these populations were not revealed. To gain a more refined view of each of the distinct cell populations, we extracted and reclustered each population.

## Developmental trajectory of female germ cells

Unlike mammals, female zebrafish can produce new oocytes throughout their life span due to the presence of GSCs (*Beer and Draper, 2013*; *Cao et al., 2019b*). Zebrafish therefore provide a unique opportunity to study female GSCs in a vertebrate. The genes that regulate GSC maintenance or progression of progenitor cells toward differentiation are not well defined in zebrafish. To gain further

insight into genes that regulate germ cell development, we extracted and reclustered the germ cell population (*Figure 2A*) and the differentiation trajectory was immediately evident in the UMAP. To determine the organization of the cells within the UMAP, we first asked which cells express known stage-specific germ cell markers. Using *nanos2, DNA meiotic recombinase 1* (*dmc1*), and *zona pellucida protein 3b* (*zp3b*), which are expressed in GSCs, early meiotic germ cells (stage 1A pre-follicle phase), and early oocytes (stage 1B follicle phase), respectively (*Beer and Draper, 2013*; *Onichtchouk et al., 2003*; *Selman et al., 1993*; *Yoshida et al., 1998*), we found that *nanos2*-expression cells clustered to the left end of the UMAP, *dmc1*-expressing cells were in an intermediate location, and *zp3b*-expression cells clustered to the right end of the UMAP. Thus, the cells are organized into what appears to be a developmental trajectory (*Figure 2A and B*). Because our library preparation method excluded cells larger than 40 µm in diameter, it is likely that only stage IA (pre-follicle stage) or early stage IB (follicle stage) oocytes are present in our dataset (*Selman et al., 1993*). To further assess this developmental trajectory, we performed a dedicated trajectory analysis and inferred pseudotime using Monocle 3 (*Cao et al., 2019a*). We found that the directionality and the sequential gene expression along pseudotime precisely correlated with the trajectory determined by our initial germ cell recluster analysis (*Figure 2—figure supplement 1*).

With the developmental trajectory of the germ cells constructed, we were able to identify genes with enriched expression in subsets of germ cells (*Supplementary file 2*). Notably missing from the stage-specific genes listed above are genes expressed in the proliferating oocyte progenitor cells, a population that is intermediate between the GSC and germ cells that have entered meiosis (subcluster 2 in *Figure 2A*). In most organisms, oocyte progenitor cells undergo several rounds of mitotic amplifying division before entering meiosis (*Pepling et al., 1999*). A common feature of oocyte progenitor cells, including those in zebrafish ovaries, is that they divide with complete nuclear division but incomplete cytoplasmic division, thus creating a multicell cyst with synchronized developmental progression (*Marlow and Mullins, 2008*; *Pepling et al., 1999*). To identify novel markers of oocyte progenitor cells, we searched for genes with enriched expression in this subpopulation (subcluster 2 in *Figure 2A*; *Supplementary file 2*). Of the top 100 enriched genes, we found only one gene whose expression was both germ cell-specific and restricted to this subpopulation, an unannotated gene, called *zgc:194189* (*Supplementary file 2*). Further sequence analysis revealed that this gene is the zebrafish ortholog of *forkhead-box protein L2 like* (*foxl2l*; *Figure 2—figure supplement 2*; *Ruzicka et al., 2019*). FoxL2l is a paralog of FoxL2 and is present in most teleost genomes as well as in sharks, coelacanths, and spotted gar, but is absent in the genomes of land-dwelling vertebrates (*Figure 2—figure supplement 2*). Interestingly, medaka *foxl2l* (formerly *foxl3*; *Figure 2—figure supplement 2*) is expressed in oocyte progenitors and is required for progenitor cells to commit to the oocyte-fate (*Nishimura et al., 2015*). The *foxl2l*-positive cells in our dataset also express higher levels of *proliferating cell nuclear antigen* (*pcna*) relative to the GSC population (*Figure 2B and C*), suggesting that they divide more rapidly than GSCs, providing further support that this subpopulation are zebrafish oocyte progenitors.

To test the hypothesis that *foxl2l*-expressing cells are oocyte progenitor cells, we used HCR RNA-FISH to determine where these cells are located in the 40 dpf ovary. Specifically, we determined the location of subcluster 2 cells relative to GSCs (subcluster 1) and to cells that are initiating meiosis (subcluster 3). We identified GSCs using *nanos2* expression and early meiotic cells using *meiotic recombination protein 8a* (*rec8a*) expression. *nanos2* encodes an RNA-binding protein that is expressed specifically in GSCs (*Beer and Draper, 2013*) while *rec8a* encodes a meiosis-specific member of the Rad21 cohesin family that is loaded onto chromosomes during premeiotic S-phase (*Crespo et al., 2019*). Our trajectory analysis indicated that *rec8a*-expressing cells would be more developmentally advanced than those that express *foxl2l*, as would be expected given its suspected role in premeiotic S-phase (*Figure 2B and C*). Indeed, we found that *foxl2l*- and *rec8a*-expressing cells localize within multicell clusters (*Figure 2E–H*), consistent with our hypothesis that *foxl2l* is expressed in oocyte progenitors, whereas *nanos2*-expressing cells are found predominantly as single cells or doublets, as previously shown (*Figure 2D–G*; *Beer and Draper, 2013*). We found cells expressing *nanos2*, *foxl2l,* and *rec8a* often clustered near one another (*Figure 2G*). Also consistent with our hypothesis, *foxl2l*-positive cells were found as clusters of ≥4 cells and *rec8a*-positive cells were found in clusters of ≥8 cells (*Figure 2I*). Interestingly, we rarely found cells that were double-positive for *nanos2* and *foxl2l*, and those that were had minimal expression of both genes, suggesting that these

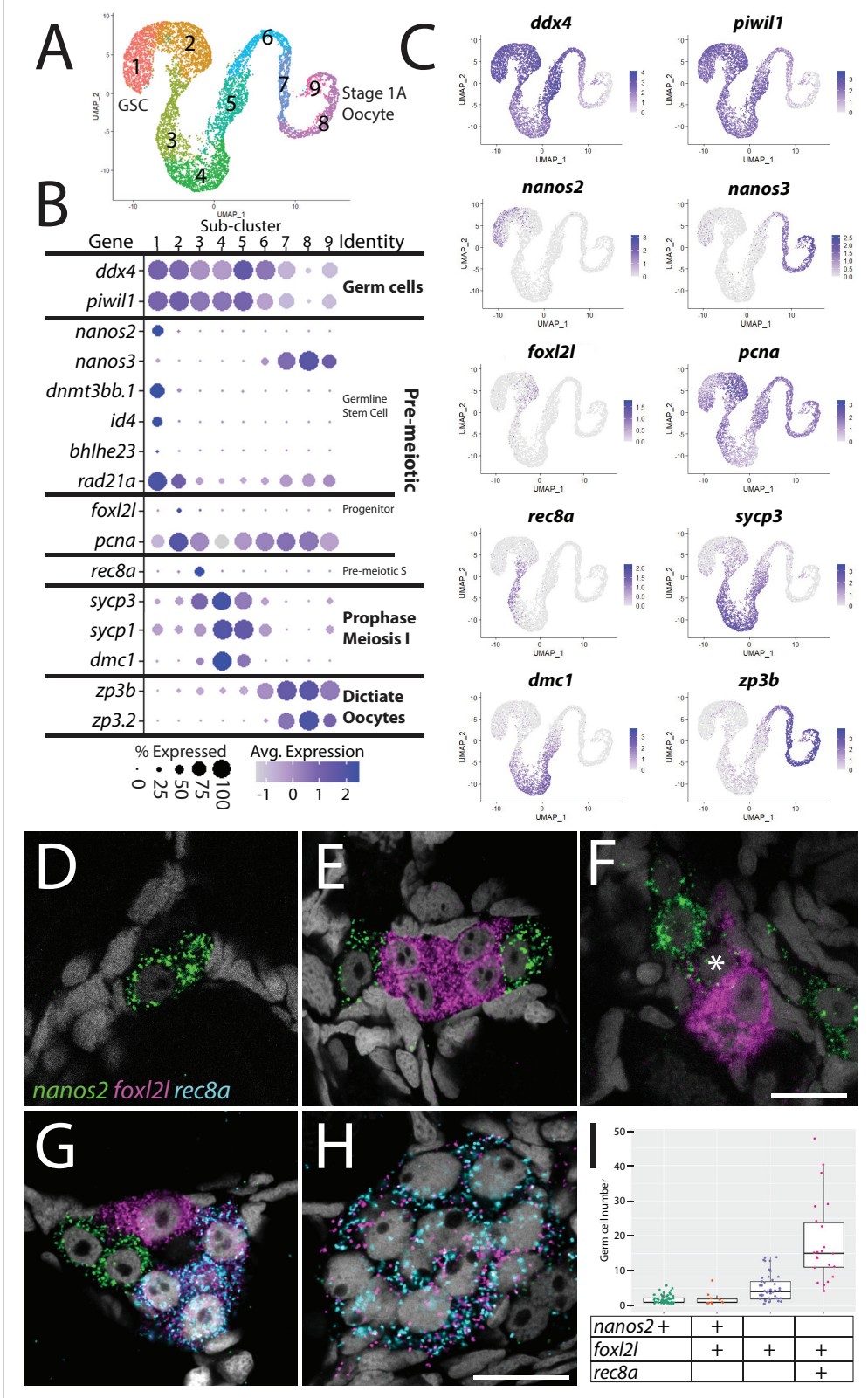

**Figure 2.** Germ cell subcluster analysis reveals developmental transitions of early germ cells. (**A**) Germ cell subcluster uniform manifold approximation and projection (UMAP) plot, with cells color-coded by computationally determined cell subtypes. (**B**) Dot plot showing the relative expression of select genes in the germ cell subclusters. Some genes, like *ddx4* and *piwil1*, are expressed in all germ cells, while others, such as *nanos2* or *rec8a*, are only

*Figure 2 continued on next page*

*Figure 2 continued*

expressed in distinct subclusters. (**C**) Gene expression UMAP plots of select genes. (**D–H**) Triple hybridization chain reaction RNA fluorescent in situ hybridization (HCR RNA-FISH) for *nanos2* (green), *foxl2l* (magenta), and *rec8a* (blue) in 40-day-old zebrafish whole-mount ovary. Asterisk in (**F**) indicates a cell double-positive for *nanos2* and *foxl2l*. (**I**) Cell number quantification of individual cysts that express the genes indicated on Y-axis. n = 70, N = 3. Scale bar in (**F**), for (**D–G**) 10 μm; in (**H**) 10 μm.

The online version of this article includes the following figure supplement(s) for figure 2:

**Figure supplement 1.** Developmental trajectory analysis of the germ cell library using Monocle 3 produced a similar trajectory to that apparent in the SURAT-based cluster analysis.

**Figure supplement 2.** Phylogenetic analysis of Foxl2, Foxl2l, Foxl3, and Foxl1 proteins.

**Figure supplement 3.** Expression of *nanos* and *pumilio* orthologs in zebrafish germ cells.

**Figure supplement 4.** Germ cell library statistics and novel zebrafish germline stem cell (GSC) markers.

**Figure supplement 5.** Gene module analysis and motif enrichment identify putative germline stem cell (GSC)-specific transcription factors.

cells were transitioning from GSC to oocyte progenitor (*Figure 2F*). Cells expressing higher levels of *foxl2l* could be divided into two populations, based on morphology and gene expression. The first consisted of cells within smaller clusters (2–4 cells) that expressed high levels of *foxl2l*, but not *rec8a*. These cells are likely early oocyte progenitors (*Figure 2E and G*). The second consisted of cells within larger clusters (≥4 cells), expressed relatively less *foxl2l* but also expressed *rec8a*. These cells were likely late oocyte progenitors, including those that are undergoing premeiotic S-phase (*Figure 2G–I*). This latter result is consistent with recent studies in medaka that found *rec8a* is a direct downstream

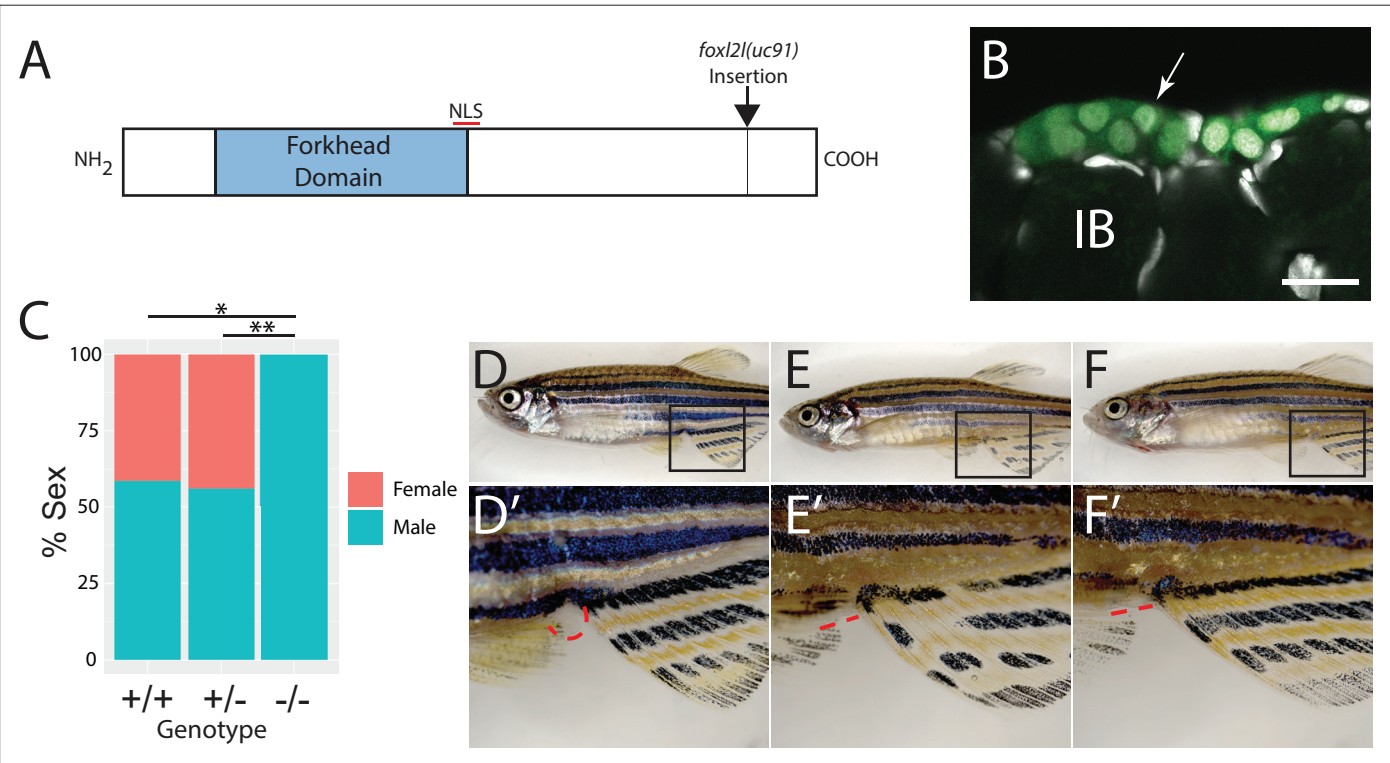

**Figure 3.** Mutational analysis of *foxl2l*. (**A**) Schematic diagram of the Foxl2l protein showing the DNA-binding forkhead homology domain (blue), the location of the nuclear localization signal (NLS), and the viral-2A-*egfp* insertion site in the *foxl2l(uc91)* allele. (**B**) GFP expression in germ cells from a *foxl2l(uc91)* knock-in allele heterozygote recapitulates endogenous *foxl2l* expression (compare to *Figure 2E and F*). (**C**) Sex ratios of *foxl2l(uc91)* heterozygotes and homozygotes. (**D–F**) Representative light micrographs of fish examined in (**C**) (n = 274, N = 4, *p=1 × 10⁻⁷, **p=2 × 10⁻⁷). Wild-type adult female zebrafish (**D**) has characteristic light-yellow pigmentation on ventral belly and a prominent anal papilla (highlighted with red dashed lines) (**D'**). (**E**) Wild-type adult male zebrafish (**E**) has dark yellow pigmentation on ventral belly and lacks an anal papilla (highlighted with red dashed lines) (**E'**). *foxl2l(uc91)* homozygous mutant (**F**) is phenotypically male. IB, stage IB oocyte.

target of *foxl2l* (*Kikuchi et al., 2020*). Thus, we identified gene expression patterns that define the premeiotic germ cell populations in the ovary.

To determine the function of *foxl2l*, we used CRISPR/Cas9-mediated gene editing to recombine a viral-2A-eGFP insert in-frame into the *foxl2l* locus, resulting in the Tg(*foxl2l:foxl2l-2A-egfp*)*uc91* allele (*Wierson et al., 2020*). The resulting fusion protein encodes all but the last 27 of 239 amino acids of the Foxl2l protein (*Figure 3A*). We found that in heterozygous animals GFP was expressed in a subset of premeiotic germ cells in the ovary (*Figure 3B*), identical to the pattern determined using HCR RNA-FISH (*Figure 2E–H*). To assess the role of *foxl2l* in germ cell development, we crossed heterozygous parents to produce homozygous mutant offspring. We found that heterozygous knock-in animals had normal sex ratios as adults (*Figure 3C–E*). By contrast, all homozygous knock-in mutant animals were fertile males as adults, indicating that *foxl2l* is required for female development (*Figure 3C and F*). In zebrafish, the ability to produce oocytes is a requirement for female development, thus loss of oocyte production, which occurs in *foxl2l* mutants, results in all-male development (*Rodríguez-Marí et al., 2010*; *Shive et al., 2010*). This is homologous to the role of *foxl2l* in medaka, which is expressed in oocyte progenitor cells and is required for these cells to commit to oogenesis, and upon loss of *foxl2l* these cells instead commit to spermatogenesis (*Nishimura et al., 2015*).

Female GSCs have not been well characterized in any vertebrate. To date, only two genes have been shown to play a cell-autonomous role in female GSC development in zebrafish: *nanos2* and *nanos3* (*Beer and Draper, 2013*; *Cao et al., 2019b*; *Draper et al., 2007*). *nanos2* and *nanos3* encode conserved RNA-binding proteins. In zebrafish, *nanos2* is expressed specifically in GSC in both male and females and loss of *nanos2* leads to sterile males (*Beer and Draper, 2013*; *Cao et al., 2019b*). By contrast, *nanos3* is only expressed in females where it plays two separate roles. First, *nanos3* (formerly called *nanos1*) is a maternally expressed gene whose mRNA localizes to primordial germ cells (PGCs) and loss of maternal function leads to loss of PGCs during early development (*Draper et al., 2007*; *Köprunner et al., 2001*). Second, *nanos3* mutant females fail to maintain oocyte production as adults due to loss of *nanos2*-expressing GSCs (*Beer and Draper, 2013*; *Draper et al., 2007*). While genetic mosaic analysis established that *nanos3* was required cell-autonomously for GSC maintenance (*Beer and Draper, 2013*), to date *nanos3* expression in premeiotic germ cells in the ovary has not been demonstrated. To address this, we identified the *nanos3*-expressing cells in our dataset. As previously reported, *nanos3* is expressed at high levels in early oocytes, but we also detected expression in the apparent GSC subpopulation, confirming that *nanos3* is expressed in GSCs as suggested by the *nanos3* mutant phenotype (*Figure 2B and C*).

In other organisms, Nanos proteins function in complex with members of the Pumilio family of RNA-binding proteins (*De Keuckelaere et al., 2018*; *Forbes and Lehmann, 1998*). The zebrafish genome encodes three orthologs of Pumilio, called Pum1–3, but it is not known which of these proteins function together with Nanos2 or Nanos3 to maintain GSC or PGC development. We found that *pum1* and *pum3*, but not *pum2*, are expressed at significant levels in GSCs (*Figure 2—figure supplement 3*). In addition, *pum1* is predominantly expressed in GSCs while *pum3* is expressed more uniformly in early oocytes (*Figure 2—figure supplement 3*). This raises the possibility that Pum1 may partner with Nanos2 while Pum3 partners with Nanos3, though it is also possible that Nanos2 could form complexes with both Pum1 and Pum3. Further mutational analysis is required to test these relationships.

We detected the highest average number of unique transcripts in GSCs (3517 unique transcripts/cell; *Figure 2—figure supplement 4*) relative to other premeiotic cells (e.g., 2315 unique transcripts/cell in progenitor cells; *Figure 2—figure supplement 4*). To identify potential regulators of GSC development, in addition to *nanos2*, we identified genes in our dataset with enriched expression in this cell population (*Supplementary file 2*). They include *inhibitor of DNA binding 4* (*id4*; *Oatley et al., 2011*), which encodes a negative regulator of transcription that is also expressed in mouse spermatogonial stem cells, *jagged canonical Notch ligand 2b* (*jag2b*; *Haddon et al., 1998*), *dickkopf WNT signaling pathway inhibitor 3b* (*dkk3b*; *Hsu et al., 2010*), *DNA methyltransferase protein 3bb.1* (*dnmt3bb.1*; *Gore et al., 2016*), *interferon regulatory factor 10* (*irf10*; *Stein et al., 2007*), and *chromogranin b* (*chgb*; *Xie et al., 2008*; *Figure 2—figure supplement 4B*). Using *id4* and *chgb* as representatives, we performed HCR RNA-FISH to verify the expression pattern in ovaries. We found that all *id4*-expressing cells also express *nanos2* (*Figure 2—figure supplement 4C*), confirming that *id4* is expressed in GSCs in the 40 dpf ovary. Interestingly, not all *nanos2*-expressing cells were *id4+*.

We obtained similar results with *cghb* (*Figure 2—figure supplement 4D*). Thus, we identified several new genes whose expression is specific to GSC. In future studies, it will be important to determine if these genes are expressed in both male and female GSCs or, like *nanos3*, are female-specific.

## Identification of germ cell stage-specific gene modules and possible transcriptional regulators

We next aimed to identify the potential transcription factors that regulate the development of germ cells. As a first step, we performed nonzero matrix factorization (NMF) to find gene modules, which are sets of genes that are co-expressed within cell clusters and subclusters and may therefore identify co-regulated genes (*Figure 2—figure supplement 5*; *Brunet et al., 2004*; *Farrell et al., 2018*; *Siebert et al., 2019*). We identified 32 gene modules (note: modules 8, 13, 19, and 29 were determined to be low-quality gene modules and therefore not considered in our analysis; see Materials and methods), and many of the gene modules showed stage-specific enrichment during germ cell development. For example, genes within modules 25, 28, 10, and 9 were specifically enriched in the GSCs, oocyte progenitors, early meiosis, and oocyte clusters, respectively (*Figure 2—figure supplement 5A and B*). We next performed gene motif enrichment analysis to find potential transcription factor-binding sites enriched within 2 kb 5′ of the transcription start site of top 20% of genes within individual modules (see Material and methods). We then verified our gene motif enrichment using the *figla* transcription factor as a test case. Folliculogenesis-specific basic helix-loop-helix (bHLH) transcription factor (Figla) is known to regulate folliculogenesis-specific gene expression in mice and zebrafish (*Liang et al., 1997*; *Qin et al., 2018*; *Soyal et al., 2000*). We found *figla* is expressed in late meiotic germ cells and early oocytes (*Figure 2—figure supplement 5C*). We searched for modules that contained genes with enriched putative Figla-binding sites and identified module 9, a module that contains genes with enriched expression in early oocytes (e.g. *zp3*; *Supplementary file 3*; *Figure 2—figure supplement 5A, C*). Therefore, we were confident that NMF and gene motif enrichment analysis could accurately identify important regulatory motifs in germ cells. To identify putative GSC-specific transcription factors, we focused our analysis on modules 7 and 25, which contained genes whose expression was specifically enriched in the GSC cluster (*Figure 2—figure supplement 5*). We found that genes within these two modules were enriched for putative binding motifs of transcription factors Bhlhe23, early growth response 4 (Egr4), and pre-B-cell leukemia homeobox 3b (Pbx3b) (*Figure 2—figure supplement 5A, D*). Importantly, GSCs also had enriched expression of Bhlhe23, Egr4, and Pbx3b (*Figure 2—figure supplement 5D*). Previous studies of *Egr4*-deficient mice have shown reduced fertility, but the activities of Bhlhe23 and Pbx3b in GSC development and function have not been reported (*Tourtellotte et al., 1999*). We performed HCR RNA-FISH to confirm the expression of *bhlhe23* and found expression specifically in a subset of *nanos2*-expressing GSCs (*Figure 2—figure supplement 5E*). Finally, the motif enrichment analysis also identified multiple genes with known functions in germ cell regulation that contain Bhlhe23 binding motifs, such as *cnot1*, which encoded a component of the CCR4/Not deadenylase complex that functions with Nanos proteins to regulate translation of select targets in GSCs (*Suzuki et al., 2012*) In addition, *dnmt3bb.1*, *tsmb4x*, and *mcm6* are expressed in GSC and have putative Bhlhe23 sites within 2 kb of their transcription start sites, suggesting that Bhlhe23 may regulate the expression of multiple genes (*Figure 2—figure supplement 5F*). The information from this analysis will allow us to build more accurate comparative gene regulatory networks between zebrafish and humans, which can be experimentally tested in the genetically tractable zebrafish.

## Identification of three follicle cell subpopulations

Follicle cells form a single-layered epithelium that encases developing oocytes once they arrest at the diplotene stage of meiotic prophase I. Teleost follicle cells are homologous to mammalian granulosa cells as evidenced by shared functions and gene expression. For example, zebrafish follicle cells express orthologs of many of the core granulosa cell-expressed genes, such as *forkhead-box protein L2* (*foxl2a* and *foxl2b*; *Crespo et al., 2013*), *anti-Mullerian hormone* (*amh*; *Rodríguez-Marí et al., 2005*), *follicle-stimulating hormone receptor* (*fshr*; *Kwok et al., 2005*), *aromatase/cytochrome P450 19a1a* (*cyp19a1a*; *Dranow et al., 2016*), and *notch receptor 3* (*notch3*; *Prasasya and Mayo, 2018*; *Figure 4B*, *Figure 4—figure supplement 1*). Oocyte and follicle cell development is coordinated through bidirectional oocyte-follicle cell-cell interactions (*Kidder and Vanderhyden, 2010*). Soon

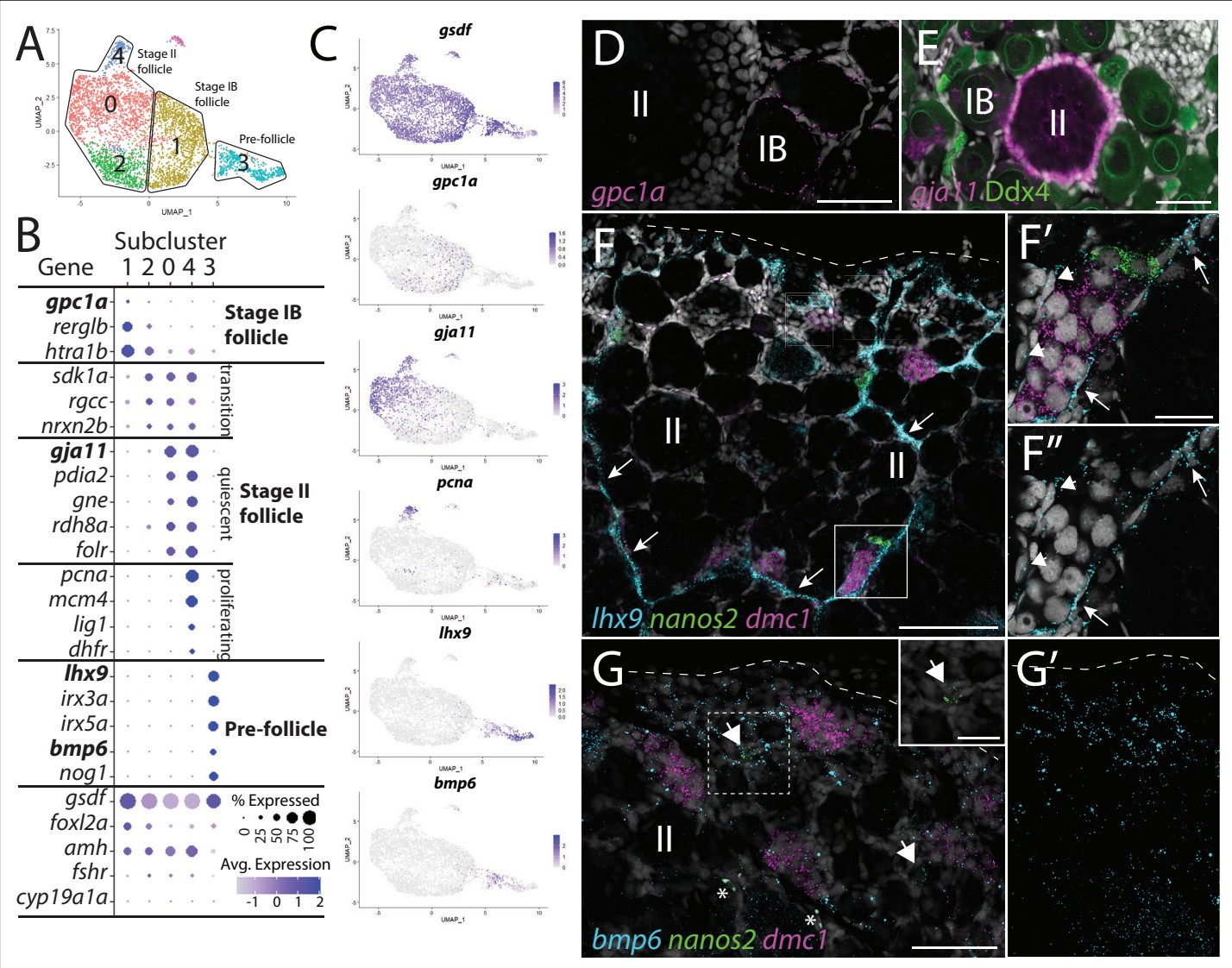

**Figure 4.** Follicle cell subcluster analysis reveals three main cell subtypes. (**A**) Follicle cell subcluster uniform manifold approximation and projection (UMAP) plot, with cells color-coded by computationally determined cell subtypes. The three main subtypes are outlined. (**B**) Dot plot showing the relative expression of select genes in the follicle cell subclusters. Some genes, like *gsdf*, are expressed in all follicle cells, while others, such as *lhx9*, are only expressed in distinct subclusters. (**C**) Gene expression UMAP plots of select genes. Cells expressing the indicated gene are colored purple, and the relative intensity indicates relative expression levels (intensity scale for each plot is on the right). (**D–G**) Hybridization chain reaction RNA fluorescent in situ hybridization (HCR RNA-FISH) on whole-mount 40-day post-fertilization (dpf) ovaries reveals the location of cell subtypes. In all panels, DNA is gray. (**D**) *gpc1a* expression (pink) is detected in follicle cells surrounding stage IB oocytes, but not stage II oocytes. (**E**) *gja11* expression (pink) is detected in follicle cells surrounding stage II oocytes, but not stage IB oocytes. Ddx4 indirect immunofluorescence (green) labels all germ cells. (**F**) Triple HCR RNA-FISH shows *lhx9*-expressing cells (blue) form tracts on the surface of the ovary (arrows) that colocalize with *nanos2* (green) and *dmc1* (pink)-expressing germline stem cells and early meiotic cells, respectively. Lateral edge of the ovary is indicated with a dashed line. (**F'**, **F"**) Higher-magnification views of regions boxed in (**F**) showing that *lhx9+* cells (arrows) surround the germ cells. (**G**) Triple HCR RNA-FISH shows *bmp6*-expressing cells (blue) are concentrated near the lateral edge of the ovary, a region that contains *nanos2* (green) and *dmc1* (pink)-expressing germline stem cells (arrows) and early meiotic cells, respectively (inset in **G** shows *nanos2* and DNA only for dashed boxed region). Asterisks identify autofluorescent red blood cells. (**G'**) *bmp6* channel only. Scale bars in (**D, E, G**) for (**G, G'**) 50 µm; (**F**) 100 µm; (**F'**) (for **F'** and **F"**) and (**G**, inset), 20 µm. IB, stage IB oocyte; II, stage II oocyte.

The online version of this article includes the following figure supplement(s) for figure 4:

**Figure supplement 1.** Gene expression uniform manifold approximation and projection (UMAP) plots of select follicle cell-enriched genes.

**Figure supplement 2.** Subcluster analysis of pre-follicle cells (*lhx9+*).

**Figure supplement 3.** Gene expression uniform manifold approximation and projection (UMAP) plots for genes that function in the retinoic acid signaling pathway.

after an oocyte has formed, it is surrounded by follicle cells and becomes dependent on these cells for factors it cannot produce, such as pyruvate, cholesterol, and select amino acids. In some cases, these compounds are transported to the oocyte via connexin-mediated gap junctions (*Su et al., 2008*). Mammalian granulosa cells are divided into two major cell types, the cumulus cells that are in direct contact with the oocyte and provide nutrients through gap junctions and the steroidogenic mural cells that form an outer layer and produce estradiol (*Gilchrist et al., 2008*). By contrast, teleost follicle cells form only a single-cell layer and are the likely source of both estradiol production and select nutrients required for oocyte development (*Devlin and Nagahama, 2002*). It can be presumed that zebrafish follicle cell maturation is coordinated with oocyte developmental progression, but little is known about stage-specific transcriptional changes that may occur in these cells. Another major difference between mammals and teleosts is that in mammals new oocytes are only formed during embryogenesis, while in many teleosts, including zebrafish, new oocytes are formed continuously from a population of GSCs. Thus, the need to produce new follicle cells extends throughout the reproductive life of the fish. However, pre-follicle cells have not been molecularly identified in any teleost (*Beer and Draper, 2013*; *Cao et al., 2019b*).

We extracted and reclustered the cells identified by *gsdf* expression to further characterize follicle cells (*Figure 4*). Because we used 40 dpf ovaries as our source of cells, the follicle cells in our dataset likely represent pre-follicle cells and those associated with stage IB and early stage II oocytes. Consistent with this, *cyp19a1a*, a gene known to be expressed in follicle cells surrounding mid-stage II and older oocytes, is only detected in a very small number of follicle cells (*Figure 4B*, *Figure 4—figure supplement 1*; *Dranow et al., 2016*; *Rodríguez-Marí et al., 2005*). We found that the majority of genes detected in follicle cells are expressed uniformly throughout the cluster (e.g., *gsdf*, *foxl2a/b*, and *amh*; *Figure 4A and B*, *Figure 4—figure supplement 1*). However, our recluster analysis distinguished five cell subclusters based on a subset of genes that had nonuniform expression (*Figure 4A* and *Supplementary file 3*). Upon further analysis, we concluded that these subclusters likely represent three developmentally relevant cell populations (e.g., subcluster 4 represents a mitotic subpopulation of otherwise developmentally similar cells in subclusters 0 and 2; *Figure 4A*).

## Stage IB follicle cells (subcluster 1)

Cells within subcluster 1 have enriched expression of the heparan sulfate proteoglycan-encoding gene, *glypican 1a* (*gpc1a*; *Gupta and Brand, 2013*), RERG/RAS-like b (*rerglb*), and *Htra serine protease 1b* (*htra1b*; *Figure 4A–C*). Using *gpc1a* as a representative gene of subcluster 1, we performed HCR RNA-FISH and determined that *gpc1a+* cells are found only surrounding stage IB oocytes (*Figure 4D*; *Selman et al., 1993*), and thus represent the primary follicle cells that first enclose oocytes to form the functional follicle. We did not observe significant expression of genes required for cell division in subcluster 1, such as *proliferating cell nuclear antigen* (*pcna*) and *dihydrofolate reductase* (*dhfr*), suggesting that stage IB follicle cells are not highly proliferative (*Figure 4B and C*).

## Stage II follicle cells (subclusters 0, 2, and 4)

Subcluster 0 contains most of the follicle cells in our dataset, which are characterized by the enriched expression of several genes, including *gap junctional protein alpha 11* (*gja11*; formerly *cx34.5*; *Mikalsen et al., 2020*; *Figure 4A–C*). To determine the location of this cell population within the ovary, we performed HCR RNA-FISH for *gja11*. These results show that follicle cells expressing *gja11* are restricted to stage II follicles (*Figure 4E*). Gene expression analysis for subcluster 2 cells suggests that these cells are a transitional state from stage IB to stage II follicle cell as they express several genes, such as *sidekick cell adhesion molecule 1a* (*sdk1a*; *Galicia et al., 2018*), which are expressed in both populations (*Figure 4B*). Gene expression analysis argues that cells within subcluster 4 are a mitotic subpopulation of subcluster 0 cells as these cells have nearly identical gene expression with cells in subcluster 0 but also express high levels of genes necessary for mitosis, such as *pcna* (*Figure 4B and C*). Follicle cell proliferation is not surprising given that the surface area of the zebrafish oocyte increases 1400-fold during oogenesis. In mammals, there is evidence that granulosa cell proliferation is regulated by pituitary-produced follicle-stimulating hormone (Fsh; *Goldenberg et al., 1972*), and consistent with this, we found the gene encoding the Fsh receptor, called *fshr*, is expressed in zebrafish follicle cells, including those in subcluster 4 (*Laan et al., 2002*; *Figure 4—figure supplement 1*). Finally, we detect only a small percentage of cells expressing *cyp19a1a* in subcluster 0, arguing that

cells within this subcluster are predominantly composed of follicle cells from early stage II oocytes, which do not express *cyp19a1a* (*Dranow et al., 2016*; *Figure 4B*, *Figure 4—figure supplement 1*).

## Pre-follicle cells (subcluster 3)

Subcluster 3 cells are part of the follicle cell lineage because they express *gsdf* (*Figure 4A–C*), yet these cells also express many genes that are unique to this subpopulation, arguing that they are distinct from the stage IB and II follicle cells identified above. Interestingly, these cells express orthologs of the core genes associated with early undifferentiated somatic gonad cells in mammals, including the transcription factors *LIM homeobox 9* (*lhx9*; *Birk et al., 2000*), *Wilms tumor 1a* (*wt1a*; *Kreidberg et al., 1993*), *empty spiracles homeobox 2* (*emx2*; *Miyamoto et al., 1997*), and *nuclear receptor 5a1b* (*nr5a1b; formerly sf1b*; *Luo et al., 1994*; *Figure 4B and C*, *Figure 4—figure supplement 2C and D*). Notably, subcluster 3 cells also uniquely express *iroquois homeobox 3a and 5a* (*irx3a and irx5a*; *Dildrop and Rüther, 2004*) whose orthologs in mammals are required for pre-granulosa cell development in the embryonic mouse ovary (*Figure 4B and C*, *Figure 4—figure supplement 2C and D*; *Fu et al., 2018*; *Kim et al., 2011*). Subcluster 3 cells do not express *gata4*, a gene required for the earliest stage of somatic gonad cell specification in mice and that is also expressed in early somatic gonad cells during the bipotential phase in zebrafish (10 dpf; *Hu et al., 2013*; *Leerberg et al., 2017*). This argues that these cells are not developmentally similar to the somatic gonad precursors present in the bipotential gonad. Finally, subcluster 3 cells express lower levels of genes associated with follicle cell differentiation and function, such as *foxl2a/b*, *amh*, *fshr,* and *notch3*, than do the stage 1B and stage II follicle cells (*Figure 4B and C*, *Figure 4—figure supplement 1*). Together, these results argue strongly that subcluster 3 cells are pre-follicle cells.

We noticed that many of the genes with enriched expression in subcluster 3 did not appear uniformly expressed in all cells within this subcluster. For example, *lhx9* appeared to have higher expression in cells distal to the subcluster 1 in the UMAP while *gsdf* and *bone morphogenetic protein 6* (*bmp6*; *Smith et al., 2006*) appeared higher in cells proximal to subcluster 1 (*Figure 4A and C*). To explore this further, we performed additional recluster analysis using only those cells derived from follicle cell subclusters 1 and 3 (*Figure 4—figure supplement 2A and B* and *Supplementary file 4*). This analysis confirmed that subcluster 3 cells can be partitioned into two distinct subclusters that, for simplicity, we have designated subclusters 3.1 and 3.2 (*Figure 4—figure supplement 2B*). While many genes, such as *emx2*, *irx3a,* and *irx5a*, appeared to be expressed uniformly in subclusters 3.1 and 3.2, our analysis confirmed that *lhx9* and *wt1a* have enriched expression in subcluster 3.1, while *gsdf* and *bmp6* appear enriched in subcluster 3.2 (*Figure 4—figure supplement 2C and D*). Another intriguing difference between these two cell populations is that subcluster 3.2 cells have enriched expression of genes that encode cell signaling components, such as *bmp6, fibroblast growth factor receptors 3 and 4* (*fgfr3* and *fgfr4*; *Sleptsova-Friedrich et al., 2001*; *Thisse et al., 1995*), and the Fgf-responsive genes *ETS variant transcription factor 4* (*etv4*; *Brown et al., 1998*) and *sprouty homolog 4* (*spry4*; *Fürthauer et al., 2001*), while subcluster 3.1 cells have enriched expression of genes that encode cell signaling attenuators, such as the Bmp ligand antagonists *noggin1 and 3* (*nog1* and *nog3*; *Fürthauer et al., 1999*), the Wnt ligand antagonist *dikkopf 1b* (*dkk1b*; *Hashimoto et al., 2000*), and the retinoic acid-degrading enzyme encoded by *cytochrome P450 26a1* (*cyp26a1*; *White et al., 1996*). In mammals, retinoic acid signaling promotes germ cells to enter meiosis (*Bowles et al., 2006*; *Koubova et al., 2006*). Therefore, it is possible that inhibition of Bmp, Wnt, and retinoic acid signaling is necessary to keep these cells in an undifferentiated, stem cell-like state, or to influence other surrounding cells.

To determine where these cells reside in the ovary, we performed HCR RNA-FISH using *lhx9* and *bmp6* as representative genes for clusters 3.1 and 3.2, respectively. Consistent with our hypothesis, we found that *lhx9+* cells do not localize around oocytes, but instead formed cord-like structures that are located on the surface of the ovary (arrows in *Figure 4F*). Triple HCR RNA-FISH for *lhx9*, GSC-expressed *nanos2,* and early meiosis-expressed *dmc1* revealed that clusters of early-stage germ cells (premeiotic and stage 1A pre-follicle phase germ cells) were always located adjacent to *lhx9+* pre-follicle cells (n = 31 clusters of *nanos2* or *dmc1*-expressing cells; N = 4; *Figure 4F, F' and F''*). This analysis also revealed that *lhx9+* cells appear to surround these early-stage germ cells (*Figure 4F' and F''*). Interestingly, there appear to be two populations of *lhx9+* cells, based on expression level and location. The first population appears to express high levels of *lhx9* and localize to the chord-like structures, which contact one side of the early-stage germ cell clusters (arrows in *Figure 4F, F' and*

F"). The second population appears to express lower levels of *lhx9* and envelop the early-stage germ cell clusters on sides that do not contact the cords (arrowheads in *Figure 4F and F"*). It is possible that the high-expressing cells correspond to the 3.1 subcluster cells while the low-expressing cells correspond to the 3.2 subcluster cells. In the medaka ovary, *sox9b*-expressing cells have been identified as pre-follicle cells and similarly formed cord-like structures on the ovarian surface that associate with early-stage germ cells, including GSCs (*Nakamura et al., 2008*; *Nakamura et al., 2010*). Indeed, we found the zebrafish *sox9b* ortholog is expressed in a subset of *lhx9* expression cells, indicating that these cells are likely functional homologs of the medaka *sox9b*-expressing cells (*Figure 4—figure supplement 2*). Finally, we determined the location of *bmp6*-expressing cells together with *nos2* and *dmc1*. *bmp6*-expressing cells also appear to colocalize to regions containing early-stage germ cells, but unlike *lhx9*, the *bmp6* staining was more diffuse, though concentrated towards the lateral edge of the ovary (*Figure 4G and G'*).

In mammals, retinoic acid signaling is required to induce germ cell entry into meiosis (*Bowles et al., 2006*; *Koubova et al., 2006*). Previous studies have argued that retinoic acid is likely produced in the zebrafish ovary by both follicle cells and interstitial cells as these cells express *aldh1a2/neckless*, which encodes the enzyme that converts retinaldehyde to retinoic acid (*Figure 4—figure supplement 3A*; *Rodríguez-Marí et al., 2013*). Given that the retinoic acid-degrading enzyme Cyp26a1 appears to be produced by the *lhx9*-expressing pre-follicle cells (*Figure 4—figure supplement 2C and D*, *Figure 4—figure supplement 3B*), and the retinoic acid receptor-encoding genes *rxrba* and *rxrbb* are expressed in premeiotic germ cells (*Figure 4—figure supplement 3C*), we speculate that the close association of premeiotic germ cells with these cells may be important for regulating GSC maintenance.

## Follicle cell-expressed *wnt9b* is required for female sexual development

In addition to their role in support and regulation of oocyte development during oogenesis, follicle cells also play key roles in promoting female sex determination and differentiation in vertebrates. Our lab previously showed that the follicle cell-expressed gene encoding the Wnt4 ligand was required for female sex differentiation, similar to the role of WNT4 in mammals (*Kossack et al., 2019*; *Vainio et al., 1999*). However, while 95% of *wnt4* mutants develop as males, we found that ~5% can develop as females. We reasoned that the partial penetrance of the phenotype could be due to the function of an additional Wnt ligand(s). To further test the usefulness of this scRNA-seq dataset for functional gene discovery, we asked if the expression of other Wnt ligands was detected in the dataset, and if so, in what cell type(s)? Our analysis confirmed that *wnt4* is expressed in follicle cells in zebrafish ovary (*Figure 5A*; *Kossack et al., 2019*). Additionally, there is significant expression of *wnt8a* (*Kelly et al., 1995*) in oocytes, *wnt9a* in stromal cells, *wnt9b* (*Crotwell and Mabee, 2007*) in follicle cells, and *wnt11* (*Makita et al., 1998*) in stromal and theca cells (*Figure 5—figure supplement 1*). We focused our attention on *wnt9b* because its apparent expression in follicle cells was strikingly similar to that of *wnt4* (*Figure 5A*). To determine if *wnt9b* was involved in sex determination and/or differentiation, we produced *wnt9b* mutants using CRISPR/Cas9. The *wnt9b(fb207)* allele contains a 57 bp deletion of exon1, which includes the splice donor sequence, and is therefore predicted to be a loss-of-function mutation (*Figure 5B*).

Domesticated zebrafish do not have sex chromosomes, instead sex is determined by a combination of genetic and environmental factors (*Kossack and Draper, 2019*). Because sex ratios in any cross can vary, it is important to compare sex ratios in mutants to their wild-type siblings. We crossed *wnt9b(fb207)* heterozygous parents and determined the genotype of their offspring at 50 dpf. We then determined the phenotype by dissecting their gonads to determine whether they had ovaries (females) or testes (males; *Figure 5C–G*). We found that wild-type animals developed similar numbers of males and females, while *wnt9b* mutants were majority male (*Figure 5H*). In addition, we found a subset of *wnt9b* mutants that had gonads that appeared to be transitioning from an ovary to a testis as they contained stage IB oocytes surrounded by germ cells undergoing early spermatogenesis (*Figure 5G*). Because oocytes are never observed in wild-type testes at 50 dpf (*Figure 5D*), the presence of oocyte in these *wnt9b* mutants suggests that these animals initiated female development before sex reversing to males. Consistent with this hypothesis, wild-type testes at 50 dpf contained germ cells at all stages of spermatogenesis, including early spermatocytes and mature spermatozoa

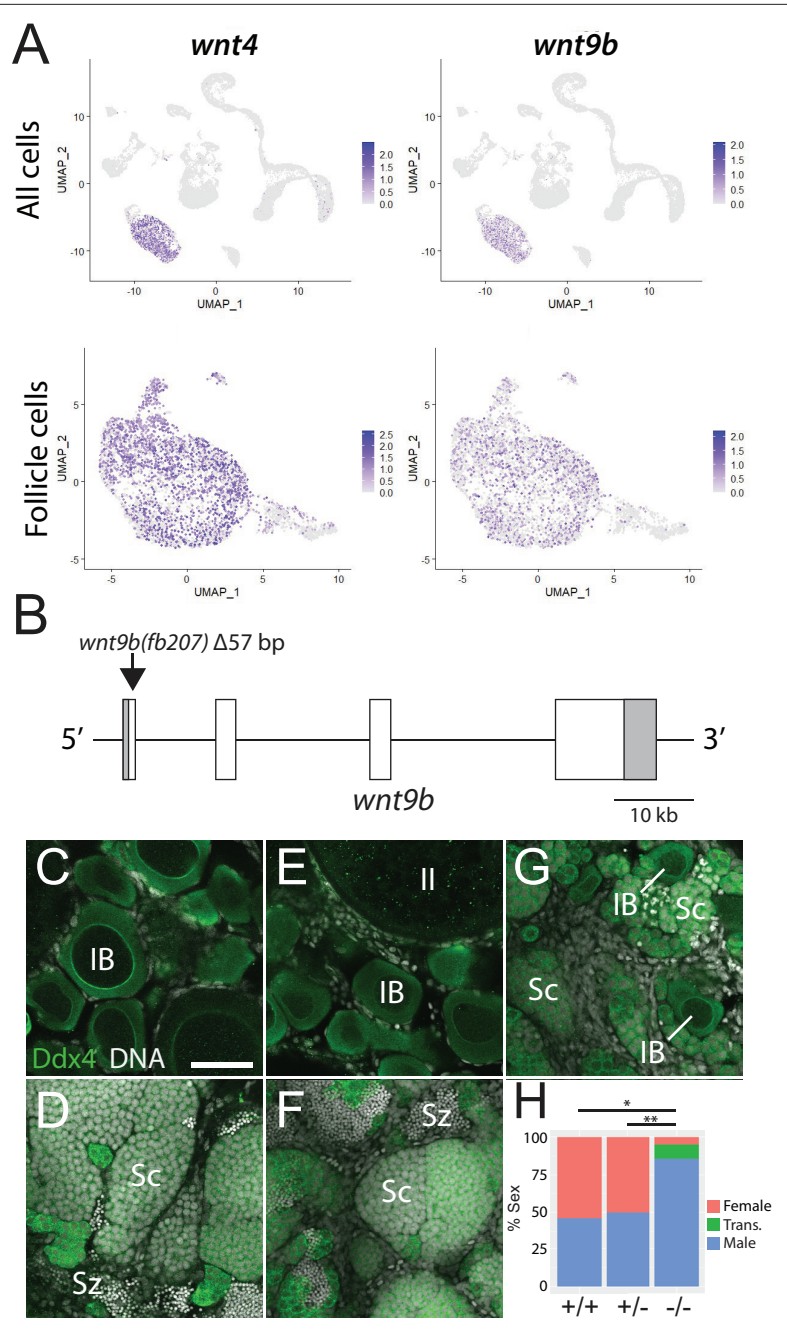

**Figure 5.** Expression and mutational analysis of *wnt9b*. (**A**) Expression plots of *wnt4a* and *wnt9b* show that *wnt9b* is expressed only in follicle cells, in a pattern nearly identical to *wnt4a*. (**B**) Schematic of the *wnt9b* genomic locus. Boxes are exons, UTR sequences are shaded. Arrow indicates approximate location of the 57 bp deletion in the *wnt9b(fb207)* allele. (**C–G**) Representative regions of gonads stained for Ddx4 protein (green) to identify germ cells. *wnt9b*(+/+) ovaries (**C**) and testis (**D**). *wnt9b*(-/-) ovary (**E**) and testis (**F**). (**G**) *wnt9b*(-/-) gonad that contains mostly germ cells that have characteristics of spermatogenesis, but also has a few stage IB oocytes. (**H**) Sex ratios of offspring produced from *wnt9b(fb207)* heterozygous (+/-) parents (n = 384, N = 3, *p=0.016, **p=0.037). Trans., transitioning; IB, Stage IB oocyte; Sc, spermatocyte; Sz, spermatozoa. Scale bar in (**C**) for (**C–G**) 50 µm.

The online version of this article includes the following figure supplement(s) for figure 5:

**Figure supplement 1.** Gene expression uniform manifold approximation and projection (UMAP) plots for Wnt ligand-encoding genes in the 40-day post-fertilization (dpf) ovary.

(*Figure 5D*), while the transitioning gonad contained only spermatocytes, but no mature spermatozoa (*Figure 5G*). The observation that some *wnt9b* mutants appear to initiate female development before sex reversing to males may indicate that the primary role of Wnt9b is in the maintenance of female sex differentiation, while Wnt4 may function primarily during sex determination.

In addition to its role in sex differentiation, we previously found that both male and female *wnt4* mutants had defects in the development of reproductive ducts that prevented release of mature gametes during spawning (*Kossack et al., 2019*). By contrast, 78.6% (n = 14) of *wnt9b* mutants were able to spawn naturally, a number that is similar to wild-type controls (100%, n = 3), arguing that *wnt9b* mutants have functional reproductive ducts. Thus, these data argue that *wnt9b* plays a similar role in female sex determination and/or maintenance to that of *wnt4*, but unlike *wnt4*, does not appear to have a role in the development of the reproductive ducts. Interestingly, *wnt9b* has previously been shown to be expressed in trout follicle cells, suggesting that it may have a similar function in sex determination and/or differentiation in other teleost (*Nicol and Guiguen, 2011*). By contrast, *Wnt9b* in mammals is not expressed in granulosa cells and there is no evidence that it plays a role in sex determination. Instead, *Wnt9b* is involved in the formation of the male and female reproductive ducts, the Wolffian and Müllerian ducts, respectively (*Carroll et al., 2005*).

## Identification of five stromal cell subpopulations

Ovarian stromal cells can be broadly defined as all ovarian cell types other than germ, follicle, or theca cells. Given this broad definition, it is no surprise that stromal cells are the least characterized cell types in the ovary. Known stromal cell types include interstitial cells, which are collagen-producing connective tissue (fibroblast) that provide structure integrity to the ovary, surface epithelial cells that surround the ovary, vasculature and associated perivascular cells, and resident blood cell types (*Kinnear et al., 2020*). To date, few of these cell types have been characterized in the zebrafish ovary. From our initial clustering, we identified a large population of cells that had enriched expression of several collagen encoding genes, such as *col1a1a* (*Fisher et al., 2003*; *Figure 1A*) and reasoned that these were likely the stromal cell population. This is further supported by the expression of *decorin* (*dcn*; *Shintani et al., 2000*), which encodes a proteoglycan component of the extracellular matrix, and *nuclear receptor subfamily 2, group F, member 2* (*nr2f2*, formerly *coup-tfII*; *Fjose et al., 1995*), both of which are expressed in mammalian ovarian stromal cells (*Figure 6A*; *Pereira et al., 1995*; *Wagner et al., 2020*). To further characterize this cell population, we performed recluster analysis and identified five probable cell subpopulations, which are described below (*Figure 6A* and *Supplementary file 5*). Of the somatic ovarian cells, the stromal cell cluster is the most transcriptionally diverse. This is also supported by Gene Ontology (GO) analysis (*Figure 6—figure supplement 1*).

### Interstitial cells (clusters 0 and 4)

Subcluster 0 is the largest of the stromal cell population and likely represents the collagen-producing fibroblast-like connective tissue (*Figure 6B*). In addition to the expression of the collagen-encoding genes, like *col1a1a*, notable genes expressed in these cells include the *chemokine (C-X-C motif) ligand 12a* (*cxcl12a*, formerly known as *sdf1a*; *Crosier et al., 2001*), the nuclear receptor *nr2f2,* the cell signaling molecule *bmp4,* and *cyp19a1a* (*Figure 6A and B*). Cells within subclusters 0 and 4 have nearly identical gene expression profiles, with the exception that subcluster 4 cells also have strong expression of genes associated with cell cycle progression, such as *pcna* (*Figure 6A and B*). Given that the ovary continues to increase in size as the fish grows, it is likely that subcluster 4 cells serve as a source for new interstitial cells similar to follicle cell subcluster 4 cells. Previous studies have shown that *cxcl12a* is essential for directing migrating PGCs to the forming gonad during embryogenesis (*Doitsidou et al., 2002*). However, we only detect expression of the *cxcl12a* receptor, *cxcr4b*, in neutrophiles and macrophage cells (*Supplementary file 1*) raising the possibility that Cxcl12/Cxcr4b signaling axis may instead be involved in recruiting these cells into the ovary (*Petit et al., 2007*). Interestingly, macrophage have been shown to play an important role in regulating spermatogenesis in the mouse testis (*DeFalco et al., 2015*). Regardless, *cxcl21a* provides a convenient marker for determining the location of this cell population in the ovary. Using FISH, we found that *cxcl12a*-expressing cells are enriched in the interstitial regions between early follicles, but not late follicles (<stage II; *Figure 6C–C'*). Together, these data argue that subclusters 4 and 0 are proliferative and nonproliferative interstitial cells, respectively.

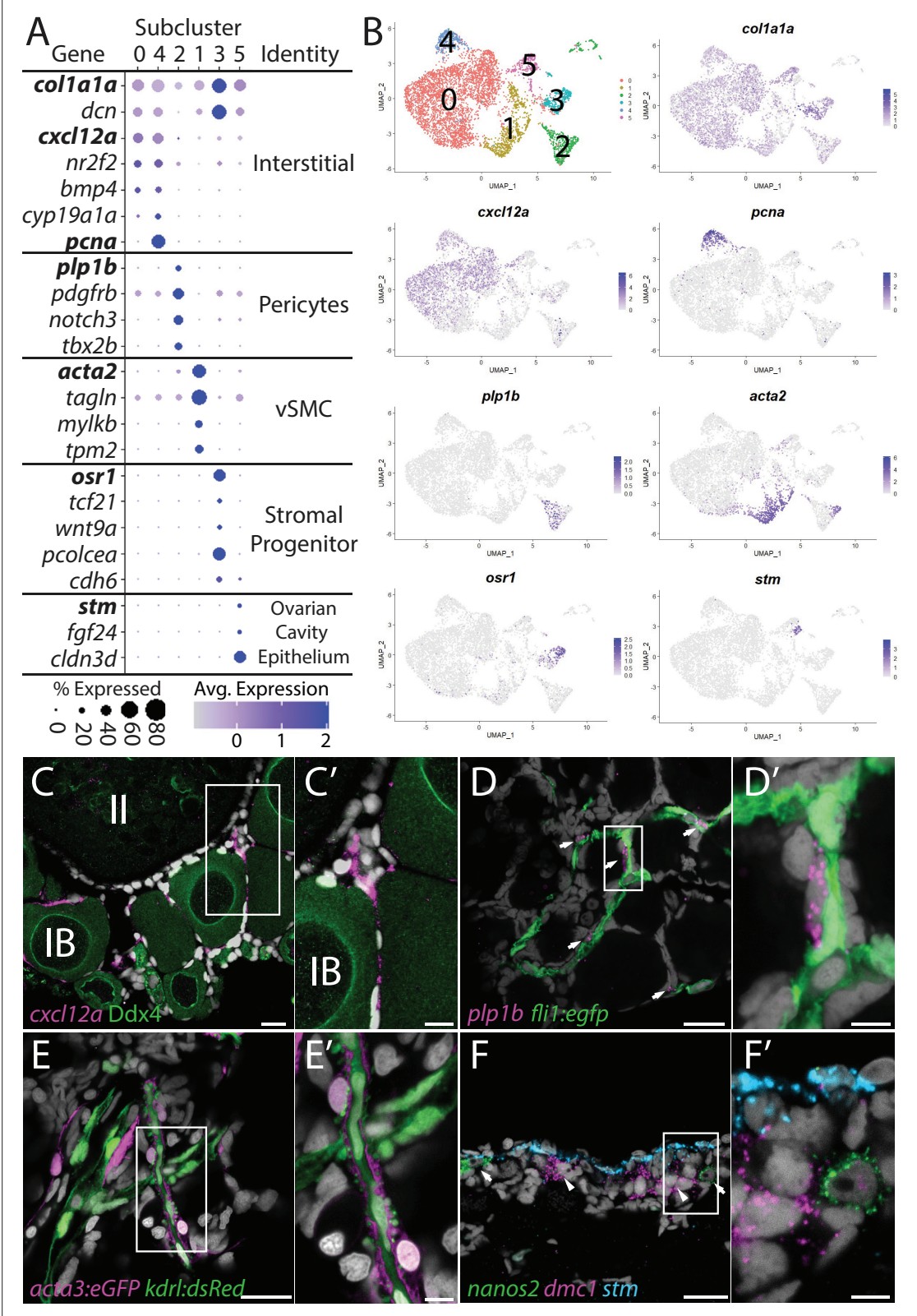

**Figure 6.** Stromal cell subcluster analysis reveals five main cell subtypes. (**A**) Dot plot showing the relative expression of select genes in the stromal cell subclusters. Some genes, like *col1a1a*, are expressed in all stromal cells, while others, such as *stm*, are only expressed in a specific subcluster. Uniform manifold approximation and projection (UMAP) plots of genes in bold are shown in (**B**). Gene expression UMAP plots of select genes. Top-left panel shows cells color-coded by computationally determined cell subtype. Cells expressing the indicated gene are colored purple, and the relative intensity

*Figure 6 continued on next page*

*Figure 6 continued*

indicates relative expression levels (intensity scale for each plot is on the right). (**C–G**) Hybridization chain reaction RNA fluorescent in situ hybridization (HCR RNA-FISH) on whole-mount 40-day post-fertilization (dpf) ovaries reveals the location of cell subtypes. In all panels, DNA is gray. (**C**) *cxcl12a*-expressing interstitial cells localize around early-stage oocytes (≤stage IB), but not around stage II oocytes. Ddx4 indirect immunofluorescence (green) labels all germ cells. (**C'**) Higher magnification of the region boxed in (**C**). (**D**) *plp1b*- expression pericytes colocalize with *fli1:egfp*-expressing blood vessels (green). (**D'**) Higher magnification of region boxed in (**D**). (**E**). *acta3:egfp* expression vascular smooth muscle cells (red) surround *kdrl:dsRed* expression blood vessels (green). (**E'**) Higher magnification of the region boxed in (**E**). (**F**) *stm* (blue)-expressing cells localize to the lateral margin of the ovary and colocalize with *nanos2* (green) and *dmc1* (pink) germline stem cells and early meiotic cells, respectively. (**F'**) Higher magnification of the region boxed in (**F**). Scale bar in (**C–E**) 20 µm; (**C'**) 10 µm, (**D', E', F'**) 5 µm. IB, stage IB oocyte; II, stage II oocyte.

The online version of this article includes the following figure supplement(s) for figure 6:

**Figure supplement 1.** Gene Ontology (GO) terms associated with the follicle and stromal cell subclusters.

**Figure supplement 2.** Stromal cell subclusters 1 and 2: pericytes and vascular smooth muscle cells.

**Figure supplement 3.** Stromal cell subcluster 3: stromal progenitor cells.

**Figure supplement 4.** Max projection view of a confocal stack showing *stm* expression (blue) and DNA (gray) in a 40-day post-fertilization (dpf) ovary.

## Perivascular mural cells: Vascular smooth muscle (subcluster 1) and pericytes (subcluster 2)

Stromal cell subclusters 1 and 2 represent the perivascular mural cells that coat the endothelial-derived blood vessels and provide stability and integrity to the vasculature. Similar to their function in the establishment of the blood–brain barrier, these cells have also been implicated in the formation of a blood–follicle barrier (**Siu and Cheng, 2012**). Pericytes are generally solitary cells associated with small diameter blood vessels (arterioles, capillaries, and venules) while vascular smooth muscle cells (vSMCs) are associated with large blood vessels where they form a continuous coating (**Gaengel et al., 2009**). In zebrafish, pericytes express *platelet-derived growth factor receptor ß* (*pdgfrb*) and *notch3*, two genes whose expression are enriched in our stromal cell subcluster 2 (**Figure 6A**, **Figure 6—figure supplement 2**; **Wang et al., 2014**). In addition, we find that subcluster 2 cells are enriched for *proteolipid protein 1b* (*plp1b*; **Brösamle and Halpern, 2002**; **Figure 6A and B**), *T-box transcription factor 2b* (*tbx2b*; **Dheen et al., 1999**), *melanoma cell adhesion molecule B* (*mcamb*; **Chan et al., 2005**), and *regulator of G protein signaling 5a and 5b* (*rgs5a* and *rgs5b*; **Wang et al., 2014**), all of which have orthologs expressed in human ovarian pericytes (**Figure 6A**, **Figure 6—figure supplement 2A**; **Wagner et al., 2020**). vSMC are characterized by expression of *α-smooth muscle actin* (*acta2*) and *transgelin* (*tagln*, formally known as SM22α; **Bahrami and Childs, 2018**). We found that both *acta2* and *tagln* are enriched in stromal cell subcluster 1 (**Figure 6A and B**, **Figure 6—figure supplement 2B** and **Supplementary file 5** ). In addition, we found that subcluster 1 has enriched expression of *myosin light chain kinase b* (*mylkb*; **Tournoij et al., 2010**), *tropomyosin 2* (*tpm2*; **Davidson et al., 2013**), and *cysteine and glycine-rich protein 1b* (*csrp1b*), further supporting the conclusion that these cells are vascular smooth muscle (**Figure 6—figure supplement 2B**).

To determine the location of pericytes in the ovary, we used *plp1b* as a marker because, in contrast to *notch3* and *pdgfrb*, our data indicate that *plp1b* is specific to pericytes. We performed HCR RNA-FISH for *plp1b* using ovaries isolated from Tg(*fli1:egfp*) that express eGFP in vascular endothelial cells (**Lawson and Weinstein, 2002**). We found that *plp1b+* cells were generally solitary, but always adjacent to blood vessels, consistent with pericytes (**Figure 6D–D'**). To determine the localization of vSMC, we imaged ovaries from Tg(*kdrl:dsRed*); Tg(*acta2:eGFP*) double transgenic animals. Tg(*kdrl:dsRed*) is expressed in vascular endothelial cells, while *acta2:egfp* is expressed in vSMC (**Kikuchi et al., 2011**; **Whitesell et al., 2014**). We found that the majority of the *acta2+* vSMC were either adjacent to or wrapped around the *kdrl+* vascular endothelial cells (**Figure 6E–E'**). Together, our data argue strongly that subclusters 2 and 1 represent pericytes and vSMC, respectively.

## Stromal progenitor cells (subcluster 3)

In addition to expressing high levels of the general stromal cell genes *col1a1a* and *dcn* (**Figure 6A**), subcluster 3 cells also express many genes that are known to play roles in early mesoderm and stromal cell development (**Figure 6A and B**, **Figure 6—figure supplement 3B and C** and **Supplementary file 5**). For example, the transcription factors *odd-skipped related 1* (*osr1*; **Tena et al., 2007**) and *transcription factor 21* (*tcf21*; **Knight et al., 2008**) are expressed specifically in stromal cell subcluster

3 cells. In mice, *Osr1* is one of the earliest genes known to be expressed in lateral plate mesoderm, derivatives of which form the kidney and gonads, and *Osr1* mutants fail to form either of these organs (*So and Danielian, 1999*; *Wang et al., 2005*). *Tcf21*-expressing cells in the mouse ovary and testis were identified through lineage labeling as multipotent mesenchymal/stromal progenitor cells that can produce all somatic cell types during development and could contribute to somatic cell turnover during aging or cell replacement following injury in the testis (*Cui et al., 2004*; *Shen et al., 2021*). Additionally, subcluster 3 cells and mouse *Tcf21*-expressing mesenchymal progenitors also share common expression of *osr1*, *procollagen C-endopeptidase enhancer a (pcolcea)*, *slit homolog 3 (slit3)*, *platelet-derived growth factor receptor a (pdgfra)*, *dcn*, *col1a1a*, *col1a2*, and *insulin-like growth factor 2 (igf2a*; *Figure 6—figure supplement 3B and C*; *Shen et al., 2021*). We therefore propose that these cells are stromal progenitor cells. As we have yet to identify where these cells reside in the ovary, we do not know if these cells are a stable population that persist into adulthood or if they are a transient population present only during the juvenile and early adult stages while the ovary is increasing in size.

## Ovarian cavity epithelium (subcluster 5)

Stromal cell subcluster 5 has enriched expression of *fgf24*, a gene that is expressed in the outer epithelial layer of the early bipotential gonad (10–12 dpf) and is required for early gonad development in zebrafish (*Leerberg et al., 2017*). Subcluster 5 cells also express the cell adhesion protein encoded by *claudin 3d (cldn3d*; formerly called *cldnc*; *Kollmar et al., 2001* #486) and *starmaker (stm)*, a gene required for otolith formation in the ear (*Figure 6A and B*; *Söllner et al., 2003*). Given the expression of *fgf24* in surface epithelial cells of the early gonad (*Leerberg et al., 2017*) and the role of claudin proteins in forming tight junctions between epithelial cells (*Tsukita et al., 2019*), it was plausible that subcluster 5 cells corresponded to the ovarian surface epithelium. To test this, we performed HCR RNA-FISH using a probe for *stm*. We chose *stm* because it is not expressed in any other cell type in the ovary and would therefore allow us to specifically identify this population of cells. We found that *stm* was specifically expressed in cells that localize to the medial and lateral edges of the 40 dpf ovary, a region that also contained *nanos2*+ GSCs and *dmc1*+ meiotic germ cells (*Figure 6F–F'*, *Figure 6—figure supplement 4*). The colocalization of *stm*-expressing somatic cells with early-stage germ cells, including GSC's, raises the interesting possibility that the lateral and medial edges of the 40 dpf ovary are the precursors to the germinal zones in the adult ovary.

To explore the relationship between the lateral and medial edges of the juvenile ovary and the germinal zone in the adult ovary, we performed HCR RNA-FISH for *stm* on transverse sections from 3-month-old adult ovaries. We found that *stm*-expressing cells localize to the membrane that forms the ovarian cavity, a fluid-filled space that forms on the dorsal side of the ovary and into which mature eggs are ovulated (*Figure 7*; *Takahashi, 1977*). We therefore designate this the ovarian cavity epithelium (OCE). The OCE is contiguous with the reproductive duct located at the posterior end of the ovary and that functions as the conduit between the OCE and genital papilla (*Kossack et al., 2019*). Interestingly, the OCE appears to adhere to the medial and lateral edges of the ovary at a location that correlates with that of the germinal zone (*Figure 7B, C and H*). To investigate this further, we prepared transverse histological sections from 3-month-old adult ovaries. We found that premeiotic and early meiotic germ cells are adjacent to where the OCE adheres to the ovary at both the lateral and medial sides (*Figure 7D–G*). Finally, we found that in addition to *stm* and *fgf24*, subcluster 5 cells also express the *fgf24*-related ligand encoded by *fgf18b*, transcription factors *paired box 2b (pax2b*; (*Pfeffer et al., 1998*) and *empty spiracles homeobox 2 (emx2*; *Morita et al., 1995*), and *wnt7aa Norton et al., 2005*; *Figure 7A*). Studies in medaka have shown that the transcription factor Pax2 is a probable direct regulator of the *stm* ortholog, *starmaker-like* (*Bajoghli et al., 2009*), raising the possibility that Pax2a may also regulate the expression of *stm* in OCE cells. Orthologs of *pax2*, *emx2*, and *wnt7aa* are expressed in the Müllerian ducts in mammals (*Roly et al., 2020*), raising the intriguing possibility that the OCE and mammalian Müllerian ducts share a common evolutionary origin. Altogether, these data argue that *stm*-expressing cells form the OCE and may also play a role in GSC niche formation and/or function.

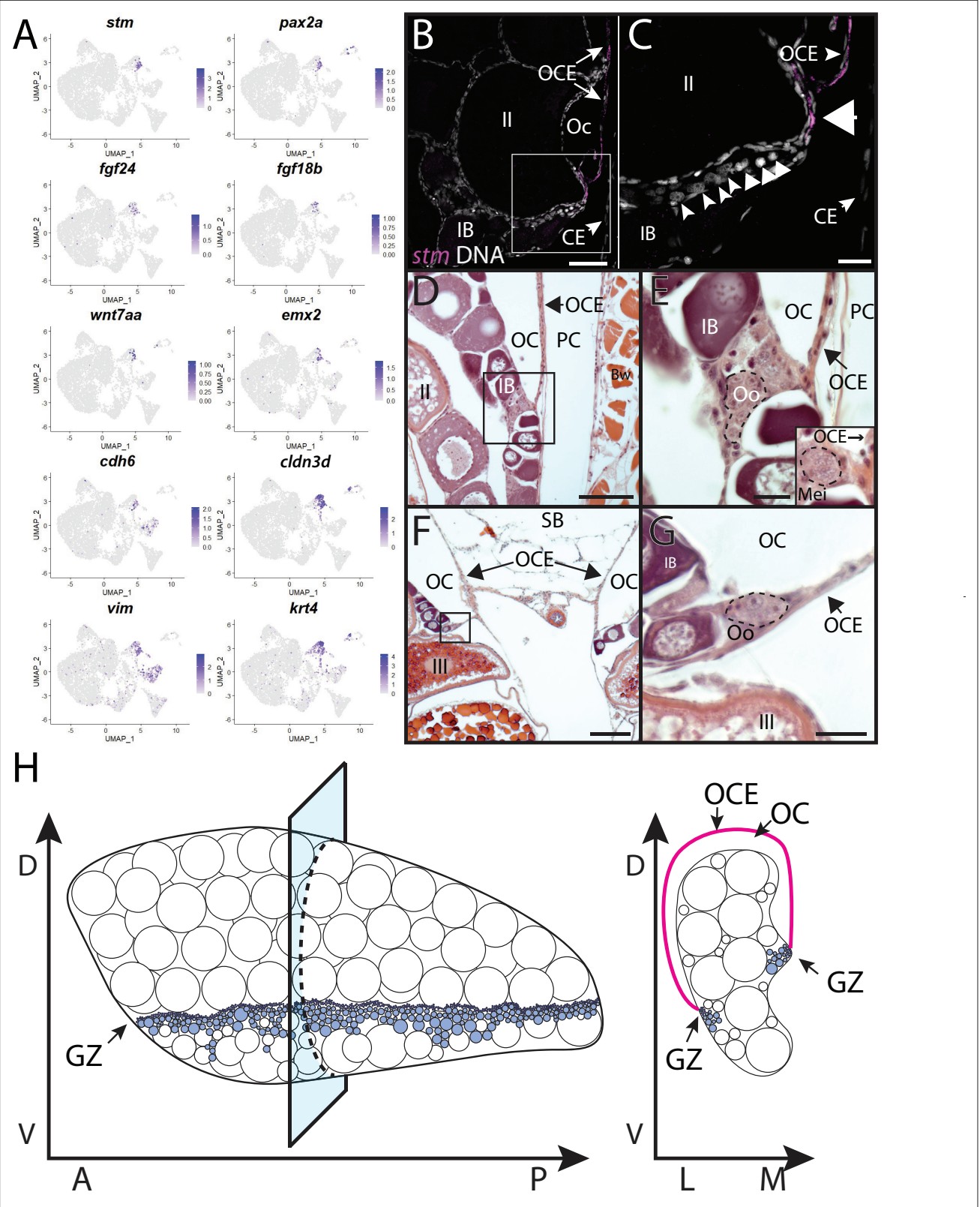

**Figure 7.** Stromal cell subcluster 3: ovarian cavity epithelium (OCE). (**A**) Gene expression uniform manifold approximation and projection (UMAP) plots for select genes whose expression is enriched in stromal cell subcluster 3. (**B**) Hybridization chain reaction RNA fluorescent in situ hybridization (HCR RNA-FISH) on transverse sections from a 3-month-old ovary showing that *stm* (magenta) is expressed in the epithelium that lines the ovarian cavity (OC). DNA is gray. (**C**) Higher magnification of region boxed in (**B**) showing that early-stage germ cells localize to the region subjacent to where the OCE

*Figure 7 continued on next page*

*Figure 7 continued*

is attached to the lateral side of the ovary (arrowhead). (**D, E**) Histological transverse sections from a 3-month-old ovary showing correlation between where the OCE attaches to the ovary at the lateral (**D, E**) and medial (**F, G**) sides, and the presence of premeiotic germ cells, characterized by large, dark staining nucleoli, and early meiotic germ cells, characterized by condensed chromosomes (inset in **E**). (**E**) and (**F**) are higher-magnification views of regions boxed in (**D**) and (**F**), respectively. PC, peritoneal cavity; CE, coelomic epithelium; SB, swim bladder; Oo, premeiotic oogonia; Mei, early meiotic germ cell; IB, stage IB oocyte; III, stage III oocyte. Scale bar in (**C, E, G**) 10 μm; (**B**) 100 μm; (**D**) 200 μm; (**F**) 250 μm.

## Theca cells and production of the 17ß-estradiol precursor, androstenedione

In addition to producing mature gametes, the other major role of the ovary is to produce the sex steroid hormone 17ß-estradiol (E2), which induces female-specific sex characteristics throughout the body. E2 is the major female sex hormone in vertebrates, including zebrafish (*Devlin and Nagahama, 2002*). In mammals, the production of E2 requires two cell types: theca and granulosa cells (*Ryan, 1979*). The primary role of theca cells is to produce the intermediates for estrogen production, using cholesterol as the starting substrate and ending with the precursor androstenedione. Androstenedione is then transferred to granulosa cells for conversion to E2 by two additional enzymes, most notably the Cyp19a1 aromatase (*Miller and Auchus, 2011*). Although many studies have investigated steroid hormone production in zebrafish, there are still significant gaps in our understanding of the biosynthetic pathway that led to E2 production. For example, while enzyme orthologs necessary for E2 production in mammals have been identified in zebrafish, the specific cell type(s) that produce E2 in zebrafish has not been determined, nor have theca cells been definitively identified. To explore

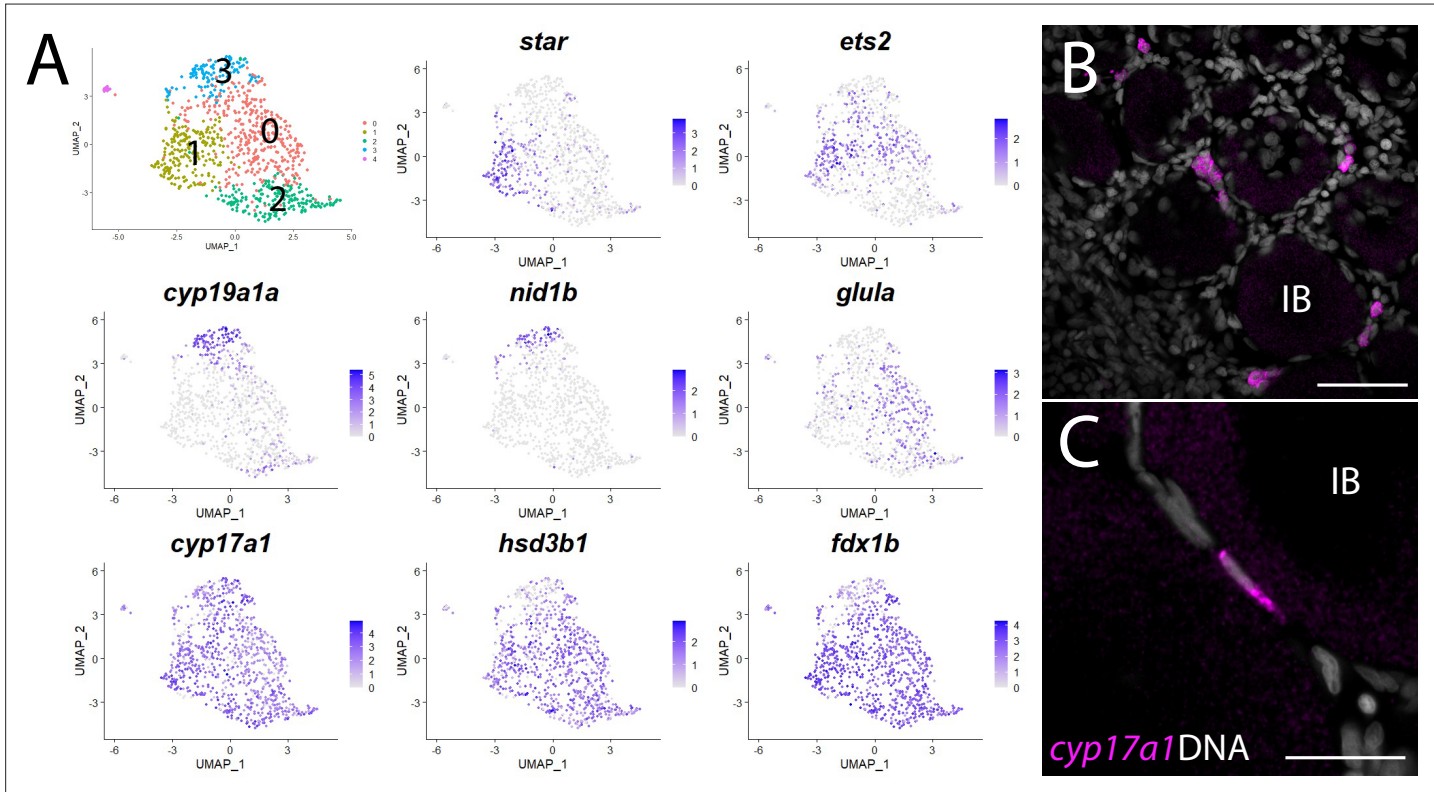

**Figure 8.** Theca cell subcluster analysis. (**A**) Gene expression uniform manifold approximation and projection (UMAP) plots of select genes. Top-left panel shows cells color-coded by computationally determined cell subtype. Cells expressing the indicated gene are colored purple, and the relative intensity indicates relative expression levels (intensity scale for each plot is on the right). (**B**) Hybridization chain reaction RNA fluorescent in situ hybridization (HCR RNA-FISH) on whole-mount 40-day post-fertilization (dpf) ovaries reveals the location of *cyp17a1*-expressing theca cells. DNA is gray. (**C**) Higher magnification of *cyp17a1*-expressing theca cells. IB, stage IB oocyte. Scale bars in (**B**) 50 μm and (**C**) 20 μm.

The online version of this article includes the following figure supplement(s) for figure 8:

**Figure supplement 1.** Expression of genes involved in theca cell development and steroid synthesis.

this further, we first extracted and reclustered the cells identified in our preliminary cluster analysis as theca cells based on their expression of *cyp11a2* (*Figures 1 and 8*).

The first and rate-limiting step in E2 synthesis in mammals is the transfer of cholesterol from the outer to inner mitochondrial membrane by steroidogenic acute regulatory protein, StAR, where it is converted to pregnenolone by the cytochrome P450 cholesterol side chain-cleaving enzyme, Cyp11a (*Miller and Auchus, 2011*). The expression of both *star* and *cyp11a1* has been detected in zebrafish ovaries (*Bauer et al., 2000*; *Ings and Van Der Kraak, 2006*). Unlike other vertebrates, teleost have two paralogs of *cyp11a1*, called *cyp11a1* and *cyp11a2*, with *cyp11a2* sharing the highest degree of similarity with other *Cyp11a1* orthologs (*Parajes et al., 2013*). Our dataset demonstrates that *cyp11a1* and *cyp11a2* are both expressed in a distinct population of somatic cells, thus identifying these as theca cells (*Figure 1*, *Figure 8—figure supplement 1*). Additionally, we found that *cyp11a2* is expressed at higher levels than *cyp11a1* and therefore may encode the primary cholesterol side chain-cleaving enzyme required for E2 synthesis in the ovary (*Figure 8—figure supplement 1*). By contrast, the majority of *cyp11a1* expression cells in the dataset are stage IA oocytes (*Figure 8—figure supplement 1*), consistent with previous results showing that in zebrafish *cyp11a1* is maternally expressed (*Parajes et al., 2013*). Finally, expression of the three remaining genes encoding enzyme orthologs required for androstenedione production by theca cells, *star*, *cyp17a1* (*Wang and Ge, 2004*) and *hydroxy-delta-5-steroid dehydrogenase, 3 beta- and steroid delta-isomerase 1* (*hsd3b1*; *Lin et al., 2015*), are also detected in theca cells (*Figure 8—figure supplement 1* and *Supplementary file 1*). As theca cells had only been previously localized by their morphology, we performed HCR RNA-FISH using *cyp17a1* to determine the location of theca cells relative to oocytes. As expected, we found that *cyp17a1*-expressing cells localize to the interstitial spaces between developing oocytes (*Figure 8B and C*). Thus, we have molecularly characterized the theca cell population in the zebrafish ovary and confirmed that, as in mammals, theca cells are the probable source of the E2 precursor, androstenedione.

Further analysis of gene expression following reclustering of only the theca cell population suggests that theca cells are not a homogeneous cell population (*Figure 8A* and *Supplementary file 6*). We found several genes that had nonuniform expression within theca cells in zebrafish, leading to the identification of four theca cell subpopulations (*Figure 8A* and *Supplementary file 6*). Examples of genes expressed nonuniformly in theca cells include *star* (subcluster 1), *cyp19a1a* and *nidogen 1b* (*nid1b*), which encodes a basement membrane-associated protein (subcluster 3), *v-ets avian erythroblastosis virus E26 oncogene homolog 2* (*ets2*; subclusters 0 and 1), and *glutamate ammonia ligase A* (*glula*), which encodes an enzyme involved in glutamine synthesis (subcluster 0; *Figure 8A*). Nonuniformity of theca cells has been previously noted in medaka based on the expression of select genes (*Nakamura et al., 2009*), providing support that some of these subpopulations have biological relevance that warrants further study. However, nonuniform expression of *star* was unexpected as StAR is thought to catalyze the rate-limiting step in steroid synthesis – the transfer of cholesterol from the outer to the inner mitochondrial membrane (*Miller and Auchus, 2011*). This may indicate that the theca cells in our dataset represent at least two developmental states: *star-* immature and *star+* mature theca cells. Alternatively, StAR may not be required for androstenedione production by theca cells and thus E2 production. Two lines of evidence support this hypothesis. First, *cyp19a1a* mutant zebrafish all develop as males, indicating that E2 production is necessary for female development (*Dranow et al., 2016*). Second, and in contrast to *cyp19a1a* mutants, *star* mutant zebrafish have a normal sex ratio but females are infertile due to defects in oocyte maturation (*Shang et al., 2019*). These results argue that StAR is not required for the synthesis of all sex steroids in zebrafish, including E2.

## Identification of the probable pathway for estrogen production in zebrafish ovary

There are two possible biosynthetic pathways for the conversion of androstenedione to E2, which we will refer to as pathways 1 and 2 (*Figure 9A*). Both pathways require two enzymatic steps, and the function of the Cyp19a1a aromatase. However, the pathways differ in the intermediate produced and at what step Cyp19a1a functions. Specifically, pathway 1 produces testosterone as an intermediate while pathway 2 produces estrone (E1; *Figure 9A*). Pathway choice is therefore not dictated solely by the expression of Cyp19a1a, but instead by the presence or absence of the two additional enzymes encoded by *17ß-hydroxysteroid dehydrogenase types 1 and 3*, called Hsd17b1 and Hsd17b3. While

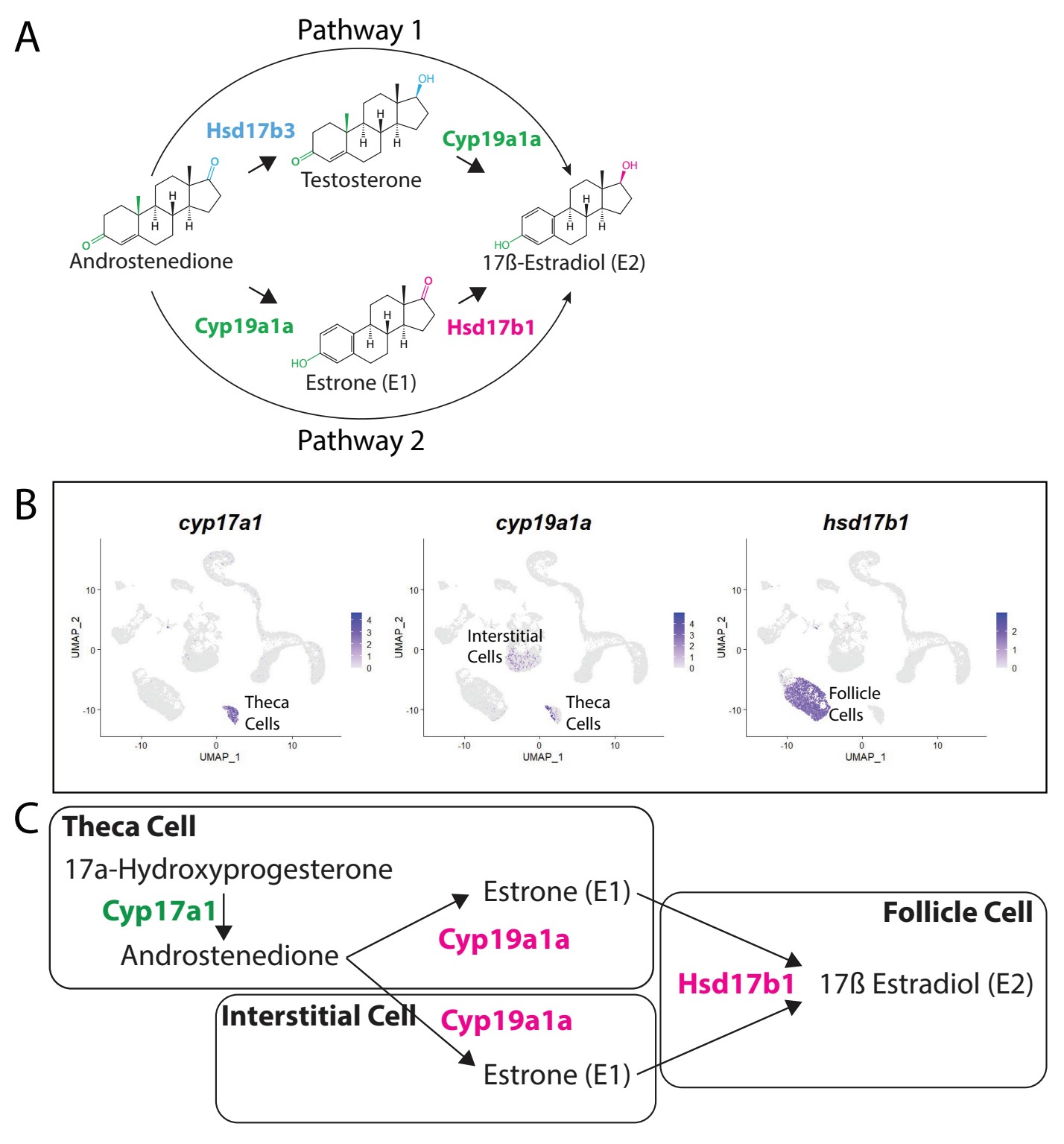

**Figure 9.** Pathway for 17ß-estradiol (E2) synthesis in the zebrafish ovary. (**A**) Two possible pathways for E2 synthesis starting with androstenedione. Colors correspond to the region of the molecules being modified and to the enzyme that catalyzes the modification. (**B**) Gene expression uniform manifold approximation and projection (UMAP) plots of select genes. Top-left panel shows cells color-coded by computationally determined cell subtype. Cells expressing the indicated gene are colored purple, and the relative intensity indicates relative expression levels (intensity scale for each plot is on the right). (**C**) Proposed pathway for E2 synthesis in the zebrafish ovary, starting with the 17a-hydroxyprogesterone intermediate precursor, together with the cell types where each reaction occurs.

*Figure 9 continued on next page*

*Figure 9 continued*

The online version of this article includes the following figure supplement(s) for figure 9:

**Figure supplement 1.** Pathway for sex hormone synthesis in the juvenile zebrafish.

these enzymes catalyze similar reactions, in zebrafish Hsd17b3 prefers androstenedione as a substrate while Hsd17b1 prefers estrone (*Mindnich et al., 2004*; *Mindnich et al., 2005*; *Figure 9A*, *Figure 9—figure supplement 1*). By contrast, current evidence argues that the aromatase Cyp19a1a does not have preference for androstenedione over testosterone (*Guiguen et al., 2010*; *Figure 9A*). It therefore follows that the expression of Hsd17b1 vs. Hsd17b3 will determine pathway preference. Though it has been proposed that testosterone is the normal intermediate for E2 production in many fish (*Devlin and Nagahama, 2002*), the pathway by which E2 is produced in the zebrafish ovary remains unknown. Thus, knowing which genes are expressed and in what cells is key to understanding how E2 production is regulated in the zebrafish ovary.

Our lab previously reported that *cyp19a1a* was likely expressed in theca cells in the early ovary and then was later upregulated in follicle cells surrounding oocytes that had progressed past mid-stage II, a stage that is underrepresented in 40 dpf ovaries (*Dranow et al., 2016*). Our scRNA-seq data verified that *cyp19a1a* is expressed in theca cells (*Figure 9B*) but not in early follicle cells (*Figure 9B*, see also *Figure 4B* and *Figure 4—figure supplement 1*). Unexpectedly we found significant expression of *cyp19a1a* in interstitial stromal cells. Thus, it is likely that three district ovarian cell populations contribute to E2 production in zebrafish: theca cells, follicle cells, and stromal cells (*Figure 9B and C*). Finally, we determined in what cell types *hsd17b1* and *hsd17b3* were expressed. We found that *hsd17b1* is expressed strongly in follicle cells while *hsd17b3* expression appears to be only weakly expressed in premeiotic germ cells (*Figure 8—figure supplement 1*). Three major conclusions can be drawn from this analysis: (1) the major pathway for E2 biosynthesis in the zebrafish ovary occurs via pathway 2, (2) Hsd17b1, not Cyp19a1a, catalyzes the final enzymatic step in E2 synthesis, and (3) follicle cells are the major source of E2 production in the zebrafish ovary (*Figure 9*).

## Conclusions

We have presented here the transcriptome of the 40 dpf zebrafish ovary at single-cell resolution, analysis of which has allowed us to define the major cell types present in the juvenile ovary. Further recluster analysis, supported by molecular and mutational evidence, revealed an unexpected level of cell subtype heterogeneity within the somatic cells that likely play important roles in regulating germ cell development as well as the development of female sexual characteristics through the production of estrogen. Our data provides strong support that orthologs of genes involved in mammalian ovary development and function likely play conserved roles in zebrafish, further validating zebrafish as a relevant model for understanding vertebrate ovarian development and function. Similarly, the major cell types required for mammalian ovary development and function have clearly identifiable homologs in the zebrafish ovary. However, unlike mammals, zebrafish females can produce new follicles throughout adult life from a population of self-renewing GSCs and pre-follicle progenitor cells, thus allowing the study of these unique cell types in a tractable vertebrate research animal. Importantly, this reference dataset will facilitate the use of zebrafish to study female reproductive diseases and disorders of sexual development.

## Materials and methods

### Key resources table

| Reagent type (species) or resource | Designation | Source or reference | Identifiers | Additional information |
|---|---|---|---|---|
| Strain, strain background (*Danio rerio*) | AB (wild-type) | University of Oregon, Eugene | ZIRC Cat ID: ZL1 | Originally obtained from the University of Oregon, Eugene |
| Genetic reagent (*D. rerio*) | Tg(*piwili1:egfp*)[uc02] | PMID:20737503 | | |
| Genetic reagent (*D. rerio*) | Tg(*fli1:egfp*)[y1] | PMID:12167406 | ZIRC Cat ID: ZL1085 | |

*Continued on next page*

*Continued*

| Reagent type (species) or resource | Designation | Source or reference | Identifiers | Additional information |
|---|---|---|---|---|
| Genetic reagent (*D. rerio*) | Tg(*kdrl:dsRed*)<sup>pd27</sup> | PMID:21397850 | ZIRC Cat ID: ZL4589 | |
| Genetic reagent (*D. rerio*) | Tg(*acta2:eGFP*)<sup>ca7</sup> | PMID:24594685 | ZIRC Cat ID: ZL9948 | |
| Genetic reagent (*D. rerio*) | *foxl2l*(*uc91*) | This paper | | CRISPR/Cas9-induced in-frame insertion of V2A-eGFP |
| Genetic reagent (*D. rerio*) | *wnt9b*(*fb209*) | This paper | | CRISPER/Cas9-induced deletion mutation |
| Gene (*D. rerio*) | *foxl2l* | PMID:28193729 | ZDB-GENE-081022-71 | |
| Gene (*D. rerio*) | *wnt9b* | PMID:17948314 | ZDB-GENE-080201-1 | |
| Recombinant DNA reagent | pGTag-*foxl2l-eGFP-ßactin* (plasmid) | This paper | | *Foxl2l* version of pGTag-eGFP-ßactin |
| Antibody | Anti-Ddx4 (chicken polyclonal) | PMID:30653507 | | IF (1:1500) |
| Antibody | Anti-Ddx4 (rabbit polyclonal) | PMID:10811828 | | IF (1:1500) |
| Chemical compound, drug | Type II collagenase | Worthington | Cat# NC9870009 | |
| Chemical compound, drug | Trypsin | Worthington | Cat# LS003708 | |
| Chemical compound, drug | Trypsin inhibitor | MP Biomedicals | Cat# 100612 | |
| Chemical compound, drug | Collagenase I | Sigma-Aldrich | Cat# C0130 | |
| Chemical compound, drug | Collagenase II | Sigma-Aldrich | Cat# C6885 | |
| Chemical compound, drug | Hyaluronidase | Sigma-Aldrich | Cat# H4272 | |
| Chemical compound, drug | 10X TrypLE (trypsin) | Thermo Fisher | Cat# A1217701 | |
| Commercial assay or kit | Chromium Single-cell 3' Library and Gel Bead kit V3 | 10X Genomics | | |
| Software, algorithm | Cell Ranger (v3.0.2) | 10X Genomics | | |
| Software, algorithm | SoupX (v0.3.1) | PMID:33367645 | | |
| Software, algorithm | DoubletFinder (v2.0.2) | PMID:30954475 | | |
| Software, algorithm | Seurat (v3.1.0) | PMID:31178118 | | |
| Software, algorithm | Monocle 3 (v0.1.3) | PMID:30787437 | | |
| Software, algorithm | MEME (v5.1.0) | PMID:25953851 | | |

## Animals

Zebrafish used for scRNAseq libraries are Tg(*piwil1:egfp*)<sup>uc02</sup> transgenics (formerly *ziwi:egfp*; *Leu and Draper, 2010*) in the AB strain. Dissected ovaries for whole-mount immunochemistry, RNA in situ hybridization, FISH, and HCR RNA FISH were using the AB strain. The following transgenic lines were also used: Tg(*fli1:egfp*)<sup>y1</sup> (*Lawson and Weinstein, 2002*); Tg(*kdrl:dsRed*)<sup>pd27</sup> (*Kikuchi et al., 2011*); and Tg(*acta2:eGFP*)<sup>ca7</sup> (*Whitesell et al., 2014*). All animal work was carried out with approval of the UC Davis IACUC.

## Zebrafish husbandry

Zebrafish husbandry was performed essentially as previously described (*Westerfield, 2000*), with the following specific details. All zebrafish used in this study were raised in a recirculating aquatic system (Aquaneering, Inc, San Diego, CA), with the following water parameters: water temperature, 28–29°C; pH, 7.0–7.2 (maintained with Sodium Bicarbonate); conductivity, 600–700 µS (maintained with Crystal Sea Marine Mix, Aquaneering, CA); ammonia, 0 ppm; nitrate, 40–60 ppm; water exchange, 10% total volume/day; water source, deionized water. At 5 dpf, zebrafish larvae are placed into 1 l aquaria containing 250 ml 4 parts per thousand (ppt) seawater (Crystal Sea), at no more than 30 fish/tank. At

8 dpf, an additional 250 ml 4 ppt seawater is added to each tank. From days 10 to 30, the tanks are maintained with a slow continuous flow of system water. From 5 to 10 dpf, larvae are feed rotifers ad libitum (*Brachionus plicatilis*, L-type; Reed Mariculture, Campbell, CA). From 10 to 12 dpf, each tank of larvae is fed both rotifers and one drop concentrated brine shrimp nauplii, twice daily (Aquaneering, Inc). From day 12 to 20 dpf, each tank of larvae is fed two drops concentrated brine shrimp nauplii, twice daily. From 20 to 30 dpf, each tank of larvae is fed three drops concentrated brine shrimp nauplii twice daily. At 30 dpf, juvenile fish are moved to 9 l tanks at 30 fish/tank with continuous system water flow. From 30 to 40 dpf, juvenile fish are fed 500 µl concentrated artemia nauplii and 100 mg Gemma Micro 300 Zebrafish Food (Skretting, France)/tank, twice daily. Starting at 40 dpf, fish are fed 100 mg Zebrafish Select Diet (Aquaneering, Inc)/tank, twice daily.

## Zebrafish ovary dissociation for single-cell RNA sequencing

### Germ cell library cell dissociation

Germ cells were dissociated from the somatic gonad using a modification of *Blokhina et al., 2019*. 40 pairs of 40 dpf ovaries were dissected from Tg(*piwil1:egfp*)*uc02* transgenic fish (*Leu and Draper, 2010*) and stored in a LoBind tube (Cat# 0030108302; Eppendorf) containing 2 ml of L15 medium (Cat# L5520; Sigma-Aldrich). The tissue was minced with small scissors into <1 mm pieces. 200 µl of 20 mg/ml of type 2 collagenase in L15 (Cat# NC9870009; Worthington) were added and incubated on an orbital rotator at 28°C for 35 min. The cell suspension was then gently passed through a 23G needle five times to breakup large cell clumps. 200 µl of 7 mg/ml trypsin (Cat# LS003708; Worthington) in L15 were added and incubated on an orbital rotator for 10 min or until a minimal amount of cell clumps was observed. The trypsin reaction was stopped by adding 500 µl of 20 mg/ml trypsin inhibitor (Cat# 100612; MP Biomedicals) in L15. The cells were centrifuge for 3 min at $300 \times g$ and the supernatant carefully removed. The cell pellet was then resuspended and washed two times with 5 ml of L15 using a P1000 pipette tip and subsequently centrifuged for 3 min at $300 \times g$. Cells were resuspended in 1 ml L15 and filtered first through a 100 µm nylon filter (Cat# 431752; Corning) and then through a 40 µm nylon filter (Cat# 431750; Corning) to remove cell clumps. The filtrate was centrifuged for 3 min at $300 \times g$ and resuspended in 1 ml of 50 mg/ml BSA (Cat# A8806; Sigma-Aldrich) in phosphate-buffered saline (PBS). Cell viability and number were determined using propidium iodine (Cat# P1304MP; Thermo Fisher) and Hoechst 33342 (Cat# H3570; Thermo Fisher) staining on a Fuchs-Rosenthal hemocytometer (Cat# DHC-F01; Incyto).

### Whole ovary library cell dissociation

Somatic ovary cells were dissociated using a modification of *Elkouby and Mullins, 2017*. 40 pairs of 40 dpf ovaries were dissected from Tg(*piwil1:egfp*) transgenic fish (*Leu and Draper, 2010*) and stored in a LoBind tube containing 2 ml of L15. Tissues were minced with microdissection scissors into <1 mm pieces. After the tissue had settled to the bottom of the tube, the media were replaced with 5 ml of digestive enzyme mixture, 3 mg/ml collagenase I (Cat# C0130; Sigma-Aldrich), 3 mg/ml collagenase II (Cat# C6885; Sigma-Aldrich), and 1.6 mg/ml hyaluronidase (Cat# H4272; Sigma-Aldrich) in L15 and incubated on an orbital rotator at room temperature. The suspension was monitored every 10 min until no or a minimal number of cell clumps were observed (~30 min). Cells were centrifuged for 3 min at $300 \times g$ and resuspended in 5 ml of 5X TrypLE (Cat.No. A1217701; Thermo Fisher) in L15 and incubated on an orbital rotator at room temperature for 15 min. The trypsin reaction was stopped by adding 500 µl of 2.8 mg/ml trypsin inhibitor in L15 and incubated on an orbital rotator at room temperature for 1 min. The cell suspension was then added to 25 ml of L15 in a 50 ml conical tube to dilute the trypsin and centrifuged for 3 min at $300 \times g$. The cell pellet was resuspended and washed two times with 5 ml of L15 using a P1000 pipette and centrifuged for 3 min at $300 \times g$. The cell pellet was then resuspended and filtered as described above.

## Isolation of germ cells by FACS

Following cell dissociation of Tg(*piwil1*:egfp) transgenic ovaries, GFP+ germ cells were sorted using a MoFlo Astrios EQ Cell Sorter (Beckman Coulter) with a 70 µm nozzle. Cells were sorted using side scatter and GFP purify to identify single germ cells.

## Single-cell RNA library preparation and sequencing

Single-cell RNA sequencing libraries were prepared by the UC Davis DNA Technologies core. Briefly, barcoded 3′ single-cell libraries were prepared from dissociated cell suspensions or sorted cells using the Chromium Single-Cell 3′ Library and Gel Bead kit V3 (10X Genomics) for sequencing according to the manufacturer's recommendations. All libraries were targeted at 10,000 cell recovery and were amplified using 11 cycles. The cDNA and library fragment size distribution were verified via micro-capillary gel electrophoresis on a Bioanalyzer 2100 (Agilent). The libraries were quantified by fluoro-metry on a Qubit instrument (Life Technologies) and by qPCR with a Kapa Library Quant kit (Kapa Biosystems) prior to sequencing. Libraries were sequenced on a HiSeq 4,000 sequencer (Illumina) with paired-end 100 bp reads.

## Cell Ranger genome reference and gene annotation file generation

A General Transfer Format (GTF) gene annotation file (release 96) for the GRCz11 zebrafish genome was downloaded from Ensembl Genome Browser and filtered using the 'mkgtf' function in Cell Ranger (v3.0.2; 10X Genomics) to retain the following attributes: protein_coding, lincRNA, and antisense. A genome reference file was generated with Cell Ranger's 'mkref' function using the GRCz11 zebrafish genome obtained from the Ensembl Genome Browser with alternative loci scaffolds removed and the filtered GTF file.

## Count file generation

GRCz11 zebrafish genome FASTQ file from Ensembl Genome Browser with alternative loci scaffolds removed and the filtered GTF file described above were used to generate the count file using the 'count' function in Cell Ranger (v3.0.2; 10X Genomics). 'Expect-cells" was set to 10,000 based on estimated cell recovery.

## Cell cluster analysis

Analysis scripts and data for the cell cluster analysis are available on GitHub (https://github.com/yulongliu68/zeb_ov_ssRNAseq, copy archived at swh:1:rev:3430147079ab3840afdb725b01652f-caeda5f78d; Liu, 2022), and the final clustering can be explored online at the Single Cell Portal (The Broad Institute of MIT and Harvard, https://singlecell.broadinstitute.org/single_cell/study/SCP928). Briefly, the expression matrices generated from Cell Ranger (germ cell library zx1_40gc, whole ovary libraries zx2_40ov and zx4_40ov) were first processed with SoupX (v0.3.1; Young and Behjati, 2020) to remove ambient RNA. During dissociation and library generation, lysed cells can lead to ambient RNA levels that interfere with subsequent analyses. Despite several washing steps post dissociation, the initial data exploration revealed the presence of oocyte RNA across a variety of cell types. SoupX is designed to identify and remove ambient RNA contaminations. We used non-oocyte cells and inferred top oocyte-specific genes to quantify the extent of the contamination.

The adjusted datasets were then processed with DoubletFinder (v2.0.2; McGinnis et al., 2019) to determine and remove doublets that are expected in any large-scale single-cell RNA-sequencing datasets. DoubletFinder uses cell expression proximity and artificially generated doublets to deter-mine potential doublets. Prior to assessing doublets, we performed quality control on the dataset. For the sorted germ cell library, we retained cells with 200–6000 genes, less than 150,000 unique tran-scripts, and less than 5% mitochondrial transcripts. For whole ovary libraries, we retained the cells with 200–8000 genes, less than 200,000 unique transcripts, and less than 20% mitochondrial transcripts. These cutoffs were determined by the gene and transcript distributions of those libraries. Blood cells and a small number of germ cells were also removed from the whole ovary libraries. DoubletFinder was run using a conservative 5% estimated cell doublet cutoff (Figure 1—figure supplement 1).

Clustering analyses were conducted using Seurat (v3.1.0; Stuart et al., 2019) and the 'SCTransform' workflow to normalize, identify variable genes, and perform scaling of the data. Principal component analysis (PCA) was performed using the top 3000 variable genes, and principal components consid-ered were chosen based on the standard deviation of the elbow plot, p-value of the jackstraw plot, and biological knowledge of the genes in individual components. UMAP analysis and plots are gener-ated based on selected principal components. Gene expression tables for all cells, germ cell subset, pre-follicle cell subset, follicle cell subset, stromal cell subset, and theca cell subset are available for downloaded at Dryad (https://doi.org/10.25338/B8FH12).

## Trajectory analysis

Monocle 3 (v0.1.3) was used for trajectory analysis (*Cao et al., 2019a*). The expression matrix was exported from the Seurat object and used as Monocle 3 input. UMAP was used for dimensionality reduction. We used GSC-specific *nanos2* expression to identify the root of the trajectory and measured pseudo-time.

## Gene module analysis and motif enrichment

Non-negative matrix factorization (NMF) as a dimensionality reduction strategy was used to identify groups of co-expressed genes as previously described (*Brunet et al., 2004*; *Farrell et al., 2018*; *Siebert et al., 2019*). Expression data for the top 3000 variable genes as identified in the Seurat analyses were used as input to the NMF analysis. To identify the optimal number of gene modules that can describe the dataset, we tested a broad range of K values from 10 to 100 in increments of 5 and then reduced to an increment of 1 in a range between 30 and 40. The optimal K was identified as 36 for the germ cell dataset. Identified gene modules were then filtered to remove low-quality modules if a reproducibility score was lower than 0.6 or the gene module consisted of less than 10 genes.

For motif enrichment within 5′ regions of co-expressed genes, we extracted 2 kb upstream of the transcription start site of the top 20% of the genes within a module based on gene scores. Gene score is a metric describing how well a particular gene reflects the expression of the associated gene module. All sequences were from Ensembl and extracted using biomaRt (v2.40.4; *Cunningham et al., 2019*). We used MEME (v5.1.0; *Bailey et al., 2015*; *McLeay and Bailey, 2010*) for motif enrichment analysis with the following parameter: "`--scoring avg --method fisher --hit-lo-fraction 0.25 --evalue-report-threshold 10.0 --control --shuffle-- --kmer 2 sequence`"; and the following databases: jolma2013.meme, JASPAR2018_CORE_vertebrates_non-redundant.meme, and uniprobe_mouse.meme.

## GO analysis

The top 25% of the cluster genes (sorted by p-value) were used for GO analysis inputs. Go enrichment was assessed using g:Profiler2 (v0.2.0; *Raudvere et al., 2019*). The zebrafish biological process database was used, and enriched GO terms with the significance of <0.05 were considered.

## RNA in situ hybridization

All ovary samples were fixed in situ with 4% paraformaldehyde overnight at 4°C, followed by dissection. Samples were dehydrated with 100% methanol for 10 min at room temperature and stored at –20°C at a minimum overnight with fresh 100% methanol. Each probe was independently assayed a minimum of two times (N ≥ 2) using five 40 dpf ovary lobes isolated from five females for each assay (n = 10).

### Single-molecule whole-mount RNA in situ hybridization

HCR RNA-FISH probes were ordered from Molecular Instruments. Accession numbers supplied to the manufacturer for designing the probes and the lot numbers for reordering are listed in the following table. Hybridization was performed following the method detailed in Molecular Instruments protocol (MI-Protocol-HCRv3-Zebrafish version 5; Molecular Instruments) with the following modification: fixed ovaries stored in methanol were rehydrated to PBS in a graded series of MeOH:PBS (2:1, 1:1, 1:2), incubating 5 min in each. Ovaries were then washed 4 × 5 min in PBSTw (PBS + 1% Tween 20) before proceeding. Ovaries were then permeabilized with proteinase K at 50 µg/ml PBSTw for 15 min. Following in situ hybridization, samples were cleared in glycerol-PBS gradient (30, 50, and 70%) for 1 hr in each, and then mounted with ProLong Diamond Antifade Mountant (Cat# P36961, Invitrogen).

## Probes used for HCR fluorescence RNA in situ hybridization

| Target gene | Organism | Lot number | Order number | Amplifier |
|---|---|---|---|---|
| nanos2 | Zebrafish (Danio rerio) | PRB905 | d6f5fd8c-b888-4bf7-b9a1-d6d95db66d24 | B2 |
| chgb | Zebrafish (Danio rerio) | PRB908 | d6f5fd8c-b888-4bf7-b9a1-d6d95db66d24 | B1 |
| id4 | Zebrafish (Danio rerio) | PRB909 | d6f5fd8c-b888-4bf7-b9a1-d6d95db66d24 | B1 |
| foxl2l (zgc:194189) | Zebrafish (Danio rerio) | PRD331 | 165f08fe-24a8-40e3-acb3-02767bf38a37 | B1 |
| lhx9 | Zebrafish (Danio rerio) | PRD333 | 165f08fe-24a8-40e3-acb3-02767bf38a37 | B3 |
| dmc1 | Zebrafish (Danio rerio) | PRD334 | 165f08fe-24a8-40e3-acb3-02767bf38a37 | B1 |
| rec8a | Zebrafish (Danio rerio) | PRE326 | 20502418-92db-49c1-801c-fc808aa3940f | B3 |
| bhlhe23 | Zebrafish (Danio rerio) | PRE327 | 20502418-92db-49c1-801c-fc808aa3940f | B1 |
| gpc1a | Zebrafish (Danio rerio) | PRF573 | 984cd5dd-9cdc-4496-94eb-c722f6c027ae | B3 |
| plp1b | Zebrafish (Danio rerio) | PRF574 | 984cd5dd-9cdc-4496-94eb-c722f6c027ae | B3 |
| cyp17a1 | Zebrafish (Danio rerio) | PRF576 | 984cd5dd-9cdc-4496-94eb-c722f6c027ae | B3 |
| stm | Zebrafish (Danio rerio) | PRF578 | 984cd5dd-9cdc-4496-94eb-c722f6c027ae | B3 |
| bmp6 | Zebrafish (Danio rerio) | PRG855 | db0a04b4-ef3d-4944-9061-690ec2d1c956 | B3 |

## Standard whole-mount fluorescent RNA in situ hybridization

RNA in situ hybridization was performed essentially as described (*Thisse and Thisse, 2008*). Template DNA for production of a *gja11* RNA in situ probe was generated by RT-PCR. mRNA was first isolated from 40 dpf wild-type AB strain ovaries using TRI reagent (Cat# T9424; Sigma-Aldrich) and then cDNAs were synthesized with RETROScript Reverse Transcription Kit (Cat# AM1710; Thermo Fisher). A *gja11*-specific template was then generated by PCR using Phusion polymerase (Cat# M0530L; New England Biolabs) and the following primer pair: fwd, 5'-CCCTGAGCAGTCTTTTCGAGCCT-3'; rev, 5'-ta attaatacgactcactataggGTGCTTAAAGCCAGGCGGTCA-3'. Reverse primers contained a T7 RNA polymerase promoter shown in the lowercase letters. An RNA in situ probe for *cxcl12a* was generated as previously described (*Knaut et al., 2005*). For both probes, DNA templates were transcribed with T7 RNA polymerase (Cat# 10881775001; Roche) to yield digoxigenin-labeled antisense RNA probes. Probes were G-50 column purified (Cat# 45-001-398; GE Healthcare), and then diluted to 0.5–2 mg/ml in the hybridization solution with 5% dextran sulfate (Cat# S4030; Thermo Fisher). Fluorescence RNA in situ hybridization procedure was performed following *Thisse and Thisse, 2008* with the following modifications: fixed ovaries were permeabilized with proteinase K at 50 µg/ml PBSTw for 15 min, and the in situs was developed with FastRed (Cat# F4648; Sigma-Aldrich).

## RNA in situ hybridization on sections

Ovaries were isolated from 3-month-old females the afternoon after spawning. We reasoned that the ovarian cavity and its associated membrane would be more pronounced when the body cavity was devoid of mature oocytes. Whole fish were fixed overnight in neutral alcoholic formalin fixative (29.5% ethanol, 10% formalin, 50 mM sodium phosphate buffer, pH 7.2). Fish were then decalcified by incubating in 0.5 M EDTA, pH 8.0, as previously described (*Moore et al., 2002*). Fish were then processed and embedded in paraffin, and 7 µm sections were cut. RNA in situ hybridization with *stm* probes was performed as described in the manufacturer's protocol (MI-Protocol-HCRv3-Zebrafish version 5; Molecular Instruments). Prior to imaging, sections were stained 1 hr with 1 µg/ml 4',6-diamidino-2-phenylindole (DAPI) in PBS, cleared in glycerol-PBS gradient (30, 50, and 70%) for 10 min each, and then mounted with ProLong Diamond Antifade Mountant.

### Histology

Ovaries were isolated from 3-month-old females the afternoon after spawning. Whole fish were fixed overnight in Bouin's fixative (Cat# 112016; Fisher Scientific). Fish were then washed 4 × 4 hr in 70% ethanol before processing for paraffin embedding and 7 µm sections cut and stained with hematoxylin

and eosin (H&E) as previously described (*Siegfried and Steinfeld, 2021*). Imagers were acquired using a Zeiss Axiophot microscope equipped with a Leica DFC 500 camera.

## Immunofluorescence staining

Whole-mount immunofluorescence staining of ovaries was performed as previously published (*Leerberg et al., 2017*). Primary chicken anti-Ddx4 antibody (*Blokhina et al., 2019*) or rabbit anti-Ddx4 antibody (*Knaut et al., 2000*) were used at 1:1500. Secondary goat anti-chicken IgG Alexa Fluor 488 (Cat# A-11039; Thermo Fisher) or goat anti-rabbit IgG Alexa Fluor 488 (Cat# A-11008; Thermo Fisher) were used at 1:300.

## Imaging

Fluorescence RNA in situ hybridization, HCR fluorescence RNA in situ hybridization, and immunohistochemical staining samples were mounted with ProLong Diamond Antifade Mountant and imaged with either Olympus FV1000 laser scanning confocal microscope or Zeiss LSM 880 Airyscan microscope. Histological images were collected using a Zeiss Axiophot microscope equipped with a Leica DFC 500 camera. Images of whole zebrafish were collected using a Leica MZ 16F dissecting microscope equipped with a Leica DFC 500 camera.

## Germ cell quantification

Numbers of early GCs expressing *nanos2*, *nanos2+ foxl2l*, *foxl2l*, or *foxl2l+ rec8a* in individual clusters were counted in Z-stack images with 1 μm steps through the entire cluster. Individual cells must have more than 20 RNA molecules/puncta after HCR fluorescence RNA in situ to be counted as an expressing cell. 70 germ cell clusters were counted from three independent experiments with three different 40 dpf ovaries. The cell count graph was generated with ggplot2 (v3.3.2).

## Foxl2l phylogenetic analysis

Phylogenetic analysis of Foxl proteins was performed using Clustal Omega with the following protein sequences: Carp (*Cyprinus carpio*): Foxl2l XP_018932368.1; Foxl3 XP_018951720.1; chicken (*Gallus gallus*): FOXL1 XP_040537252.1, FOXL2 NP_001012630.1; coelacanth (*Latimeria chalumnae*): Foxl1 XP_005995948.1, Foxl2 XP_006001344.1, Foxl2l XP_005986027.1, Foxl3 XP_005997033.1; elephant shark (*Callorhinchus milii*): Foxl1 XP_007887429.1; Foxl2l XP_007900370.2; human (*Homo sapiens*): FOXL1 NP_005241.1, FOXL2 NP_075555.1, FOXL3 NP_001361767.1; killifish (*Nothobranchius furzeri*): Foxl2l XP_015822944.1; medaka (*Oryzias latipes*): Foxl1 NP_001116391.1, Foxl2 NP_001098358.1, Foxl2l XP_004070713, Foxl3 XP_011486175.1; Mexican tetra (*Astyanax mexicanus*): Foxl1 XP_015461274.2, Foxl2a XP_007232357.2, Foxl2b XP_007241719.1, Foxl3 XP_007251255.2; mouse (*Mus musculus*): FOXL1 NP_032050.2;, FOXL2 NP_036150.1, FOXL3 NP_001182057.1; spotted gar (*Lepisosteus oculatus*): Foxl1 XP_015223442.1, Foxl2 XP_006637658.1, Foxl2l XP_015192516.1, Foxl3 XP_006637427.1; sterlet sturgeon (*Acipenser ruthenus*): Foxl2l XP_033909703.2; *Xenopus tropicalis*: Foxl1 XP_012817795.1, Foxl2 XP_004917868.1; zebrafish (*Danio rerio*): Foxl1 NP_957278.1, Foxl2a, NP_001038717.1, Foxl2b, NP_001304690.1, Foxl2l NP_001122282, Foxl3 NP_001182055.1.

## Mutant production

### Generation of foxl2l:egfp knock-in mutant

We used the GeneWeld homology-directed repair technique to generate an in-frame insertion of a viral 2A peptide-eGFP fusion cassette downstream of amino acid 222 (*Wierson et al., 2020*). A pGTag- *foxl2l*- eGFP-B-actin-targeted integration plasmid was designed using the pGTag vector series (*Wierson et al., 2020*). 48 bp *foxl2l*-specific homology arms were cloned into the pGTag-eGFP-B-actin vector. Upstream homology arm 5'-GCGGgggAAACGCTCTAGTGCCTCTGAGCGGCAT GACTCCGCCGGTGAGCCCGGGcGGAT-3'; downstream homology arm 5'-AAGCGGAAGCTCCA TCTCCACCTGCAGTTACGCGCCGCAGAACAGTCACCCcccCCG-3'. Targeted plasmid integration into the *foxl2l* locus was done using CRISPR/Cas9 using a https://crisprscan.org/-designed sgRNA: 5'-GGGAGATGGAGCTTCCGCCC-3'. Injected F0 fish were outcrossed to wild-type and a sample of F1 embryos was screened for germline transmission of precisely integrated plasmid by PCR using the following primer sets: 5'-integration primers fwd 5'–3', rev 5'–3'; 3'-integration primers fwd 5'–3', rev 5'–3'. Precise 5' integration of the construct was confirmed for the *foxl2l(uc91)* allele using Sanger

sequencing, and expression of eGFP in germ cells was confirmed at 1.5–2 months post-fertilization. Three pairs of *foxl2l*(uc91) heterozygous fish were incrossed and their offspring genotyped at 50 dpf as follows. The wild-type and knock-in alleles were identified with independent PCR assays using the following primers: Universal_fwd: 5′-GCACATCTCCAGCTACATGC-3′; wild-type_rev: 5′-CACCGA GGTTTGCCATTAGT-3′, ampicon = 217 bp; gfp_rev: 5′-CTTCTGCTTGTCGGCCATGATATAG-3′, ampl- icon = 724 bp. Following genotyping, fish were phenotyped for sex using standard criteria (*Kossack and Draper, 2019*). Significance values were determined using one-way ANOVA.

## Generation of *wnt9b* mutant

One-celled zebrafish embryos were injected with Cas9 mRNA (1.4 ng/embryo) and a gRNA (58 pg/ embryo) (5′-CGTGGGAGAGAAGGATGCAGAGG-3′; target underlined) directed against the *wnt9b* locus. The target sequence was identified by using CHOPCHOP (*Montague et al., 2014*). Target site DNA oligos were annealed and cloned into the BsaI site of pDR274 (*Hwang et al., 2013*). Vector linearized with DraI was used as template for Maxi Script T7 RNA transcription (Cat# AM1312; Thermo Fisher) to generate gRNAs that were purified on a RNeasy Plus column (Cat# 74134; QIAGEN). Muta- genesis was verified by sequencing individual injected embryo colony PCR products covering the targeted locus. Founder heterozygotes were isolated by outcrossing grown G0 fish to TuAB wild-types and genotyping F1 offspring. Primers for genomic PCR: forward, 5′-CCCCTTTAAGAAGTTGCACTG T-3′; reverse, 5′-CAATGAGGCATTTAGAGGCTTT-3′. An allele consisting of a 57 bp deletion was identi- fied, which removed the 3′ end of the first exon and 5′ end of the first intron, leading to missplicing and a frameshift leading to a premature stop codon. To determine sex ratios of *wnt9b* mutant fish, 50 dpf fish produced from *wnt9b*+/- incross were fixed in 4% paraformaldehyde-1× PBS overnight at 4°C and stored in methanol at –20°C at least overnight. Fish were rehydrated in a methanol:1× PBS gradient and washed three times in 1× PBS prior to dissection. Sex was determined by examining dissected gonads for ovary or testis characteristics. Tissue was collected from each fish during dissection for genotyping post-dissection. Genomic DNA from tissue collected during dissection was extracted and genotyped by PCR. The primers used for genotyping this allele are 5′-CCTCCCCAGGGGCTGAAA TA-3′ (forward) and 5′-CAATGAGGCATTTAGAGGCTTT-3′ (reverse). Wild-type allele yields a 238 bp product, *wnt9b*(fb209) allele yields a 181 bp product. Significance values were determined using one- way ANOVA.

## Acknowledgements

The authors wish to thank Jack Cazet for his help with NMF analysis; Members of the Juliano and Burgess labs, Blanche Capel and Jennifer McKey for thoughtful discussions; Florence Marlow and Blanche Capel for critical reading of the manuscript; Jeffery Essner for GeneWeld plasmids; UC Davis Genome Center Sequencing Core; MCB Light Microscope Facility. R01 HD-081551 and NSF/IOS- 1456737 to BWD; R35 GM133689 to CEJ; IAD and CNK were supported by NIH grants DK126021 and GM10431 to IAD; MEF and SRW were supported in part by the NIH T32 predoctoral training program in Molecular and Cellular Biology (GM-007377); MEK was supported in part by the NIH T32 predoctoral training program in Environmental Health Science (ES-0070599); SRW was supported in part by the NSF graduate research fellowship program (2036201); YL was supported in part by the UC Davis Dissertation Year Fellowship.

## Additional information

### Funding

| Funder | Grant reference number | Author |
|---|---|---|
| National Institutes of Health | R01 HD-081551 | Yulong Liu<br>Matthew E McFaul<br>Lana N Christensen<br>Bruce W Draper |

| Funder | Grant reference number | Author |
|---|---|---|
| National Science Foundation | IOS-1456737 | Michelle E Kossack<br>Lana N Christensen<br>Bruce W Draper |
| National Institutes of Health | T32 training grant ES-0070599 | Michelle E Kossack |
| National Institutes of Health | T32 training grant GM-007377 | Matthew E McFaul<br>Sydney R Wyatt |
| National Science Foundation | GRFP 2036201 | Sydney R Wyatt |
| National Institutes of Health | R35 GM133689 | Stefan Siebert<br>Celina E Juliano |
| National Institutes of Health | UC2 DK126021 | Iain A Drummond<br>Caramai N Kamei |
| National Institutes of Health | P20 GM10431 | Caramai N Kamei<br>Iain A Drummond |

The funders had no role in study design, data collection and interpretation, or the decision to submit the work for publication.

## Author contributions

Yulong Liu, Formal analysis, Investigation, Methodology, Software, Supervision, Validation, Visualization, Writing – original draft, Writing – review and editing; Michelle E Kossack, Formal analysis, Investigation, Methodology, Visualization, Writing – review and editing; Matthew E McFaul, Formal analysis, Investigation, Methodology, Writing – review and editing; Lana N Christensen, Investigation, Methodology; Stefan Siebert, Methodology, Software, Writing – review and editing; Sydney R Wyatt, Investigation, Visualization, Writing – review and editing; Caramai N Kamei, Formal analysis, Investigation, Writing – review and editing; Samuel Horst, Nayeli Arroyo, Investigation; Iain A Drummond, Celina E Juliano, Funding acquisition, Supervision, Writing – review and editing; Bruce W Draper, Conceptualization, Data curation, Formal analysis, Funding acquisition, Investigation, Methodology, Project administration, Supervision, Visualization, Writing – original draft, Writing – review and editing

## Author ORCIDs

Yulong Liu http://orcid.org/0000-0003-0775-9258
Michelle E Kossack http://orcid.org/0000-0002-3400-5104
Matthew E McFaul http://orcid.org/0000-0002-5847-2938
Sydney R Wyatt http://orcid.org/0000-0003-2245-7813
Iain A Drummond http://orcid.org/0000-0003-3734-1231
Celina E Juliano http://orcid.org/0000-0003-4222-0987
Bruce W Draper http://orcid.org/0000-0002-4397-7749

## Ethics

This study was performed in strict accordance with the recommendations in the Guide for the Care and Use of Laboratory Animals of the National Institutes of Health. All of the animals were handled according to approved institutional animal care and use committee (IACUC) protocols (#20200 and #20201) of the University of California, Davis.

## Decision letter and Author response

Decision letter https://doi.org/10.7554/eLife.76014.sa1
Author response https://doi.org/10.7554/eLife.76014.sa2

# Additional files

## Supplementary files

- Supplementary file 1. Gene expression tables from the whole dataset.
- Supplementary file 2. Gene expression tables from reclustered germ cells.
- Supplementary file 3. Gene expression tables from the reclustered follicle cells.

• Supplementary file 4. Gene expression tables from the reclustered stage 1A follicle cells and pre-follicle cells.
• Supplementary file 5. Gene expression tables from the reclustered stromal cells.
• Supplementary file 6. Gene expression tables from the reclustered theca cells.
• Transparent reporting form

## Data availability

The raw and processed data reported in this paper are archived at NCBI GEO https://www.ncbi.nlm.nih.gov/geo/query/acc.cgi?acc=GSE191137 and in an interactively browsable forms at the Broad Institute Single-Cell Portal (https://singlecell.broadinstitute.org/single_cell/study/SCP928/40dpf-ovary-all-cells). Analysis code is archived at github (https://github.com/yulongliu68/zeb_ov_ssRNAseq, copy archived at swh:1:rev:3430147079ab3840afdb725b01652fcaeda5f78d). Gene expression tables for the cell clusters identified and the analysis R objects are archived at Dryad: (https://doi.org/10.25338/B8FH12).

The following datasets were generated:

| Author(s) | Year | Dataset title | Dataset URL | Database and Identifier |
|---|---|---|---|---|
| Liu Y, Kossack ME, McFaul ME, Christensen L, Siebert S, Wyatt SR, Kamei C, Horst S, Arroyo N, Drummond I, Juliano CE, Draper BW | 2021 | Single-cell transcriptome reveals insights into the development and function of the zebrafish ovary | https://www.ncbi.nlm.nih.gov/geo/query/acc.cgi?acc=GSE191137 | NCBI Gene Expression Omnibus, GSE191137 |
| Liu Y, Kossack ME, McFaul ME, Christensen L, Siebert S, Wyatt SR, Kamei C, Horst S, Arroyo N, Drummond I, Juliano CE, Draper BW | 2021 | Single-cell transcriptome reveals insights into the development and function of the zebrafish ovary | https://singlecell.broadinstitute.org/single_cell/study/SCP928/40dpf-ovary-all-cells | Broad Institute Single Cell Portal, SCP928/40dpf-ovary-all-cells |
| Liu Y, Kossack ME, McFaul ME, Christensen L, Siebert S, Wyatt SR, Kamei C, Horst S, Arroyo N, Drummond I, Juliano CE, Draper BW | 2021 | Single-cell transcriptome reveals insights into the development and function of the zebrafish ovary | https://doi.org/10.25338/B8FH12 | Dryad Digital Repository, 10.5061/dryad.25338/B8FH12 |

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
