## [Editor Report]

This single-cell transcriptomic analysis of young adult zebrafish ovaries provides important new data to understand gene expression patterns in numerous ovarian cell types that lead to insights into how ovary development works, and most of the principles will likely apply across vertebrates. The work will interest researchers who study gonad development, sex determination, differences (or ‘disorders’) in sex development, and impacts of the environment (including toxic pollutants) on gonad development and function.

---

## [Decision Letter]

**Decision letter after peer review:**

Thank you for submitting your article "Single cell transcriptome reveals insights into the development and function of the zebrafish ovary" for consideration by *eLife*. Your article has been reviewed by 3 peer reviewers, one of whom is a member of our Board of Reviewing Editors, and the evaluation has been overseen by Didier Stainier as the Senior Editor. The following individual involved in the review of your submission has agreed to reveal their identity: David Pépin (Reviewer #2).

Essential revisions:

1) Reduce redundancies between the results and Discussion sections. This could be achieved by using a format of 'Results and Discussion' and a short 'Conclusions' sections, rather than the current format.

2) Attend to detailed comments on the presentation.

*Reviewer #1 (Recommendations for the authors):*

Some comments on presentation, typos, etc.

P2l6 Would be helpful to the reader to have the new genes that are KOd listed in the abstract and will make the paper easier to find in an online search.

P4l24 add ref.

P10l1 something is missing here.

P12l6 Medaka's work on germ cell stem cells should be referred to here. DOI:10.1126/science.1185473.

P13l1 should be 'First, nanos3 (FORMERLY called nanos1) is a …'

P23l22 These papers say that the receptor is cxcr4b, not cxcr4a. Erez Raz and Michal Reichman-Fried Attraction rules: germ cell migration in zebrafish. Bijan Boldajipour Cxcl12 evolution – subfunctionalization of a ligand through altered interaction with the chemokine receptor.

Figure 7B, C. on the panel, indicate that magenta is stm, to match the other figures.

Figure 8. Text and legend need to say what cells formed the basis of the cells selected for subclustering.

P35l18,19 The paper has several subclusterings, so here and elsewhere indicate which subclustering is referred to.

P35l20 change 'per-follicle' to 'pre-follicle'.

P36l20 check again cxcr4a vs b.

P38l14 "…mechanisms…are…"

P38l27 Wtn4a◊Wnt4a

P38l30 '…in other teleostS.'

P42 l21 check: '40 pairs of 40 dpf ovaries were dissected from Tg(piwil1:egfp) transgenic fish (Leu and Draper, 2010) were dissected and stored'.

Fig6 sup2 'subclsuter'.

*Reviewer #2 (Recommendations for the authors):*

The manuscript is very well written and comprehensive, almost encyclopedic. However, the Results section is much too interpretative. This makes the Discussion section repetitive, as it mainly recapitulates points already made in the results. The lengthy results would be more concise if some of those points were addressed only in the discussion and this would greatly improve the legibility of the manuscript. Furthermore, it would be interesting to make parallels with other scRNAseq datasets from other organisms in the discussion. While some of the cell types like the oocyte progenitors may be unique to zebrafish, others, including stromal cell and theca cell subtypes closely parallel subtypes found in mammals which reinforces the value of the zebrafish as a model for the ovary.

P8 line 10: There is a relatively small increase in PCNA expression in the putative oocyte progenitor cells. Do other markers of mitosis display the same pattern?

Figure 2 I should read Nanos2, the gene name is clipped in the figure.

P20 line 20: What was attempted to identify the position of stromal progenitor in the ovary. What is the relevance of CD34 expression in these cells?

P21 line21. Mullerian duct epithelium and ovarian surface epithelium share a common embryonic origin in mammals. Were there further transcriptional similarities between OCE and mammalian OSE markers. What was the pattern of expression of AMHR2 in the zebrafish, which in mammals is expressed in both OSE and granulosa cells?

P23 line15. It is peculiar that only a small subset of theca cells express StAR which would be required for the initial steps of steroid biosynthesis. Are they associated with more mature follicles?

The primary data, the code for its analysis, and the reagents employed were all clearly described and available as required by *eLife*'s policies.

*Reviewer #3 (Recommendations for the authors):*

Text page 13, line 30, "Remarkably, we found that early-stage germ cells were always contacting lhx9+ pre-follicle cells". This is a bit confusing, as not all early-stage germ cells as seen in Figure 4 are in contact with the lhx9-expressing follicle cells. It seems that lhx9 follicle cells are located only on one side of a germ cell cluster. Please modify the text to make more clear.

A number of the same points are made in the Discussion as in the Results section, including, for example, the conclusions made about E2 production. In the interest of shortening the paper and reducing redundancy, Results and Discussion could be combined into one section, with a final "Conclusion" section at the end for any additional points not discussed in the 'Results and Discussion" section. See also the point below.

Overall, this is a very long manuscript and might benefit by splitting it into 2 back to back papers to make it more digestible.

Abstract, Page 2, line3 "Our somatic cell data represents all known somatic cell types, including follicle cells, theca cells and interstitial stromal cells".

It is a little confusing because the authors discuss the fibroblast-like interstitial stromal cells as one cell type of the stromal cells. In the abstract do the authors want to point out this subpopulation or might "ovarian stromal cells" be the more broad statement for the abstract here.

Introduction, Page 4, line23-24

"E2 production also requires steroidogenic theca cells which produce androstenedione, the precursor that follicle cells use for E2 synthesis (ref)." Reference is missing.

Result, Page6, line9 (26), Tables. I couldn't find the Tables.

Results, Page7, line21-22

"We found that the directionality and the sequential gene expression along pseudotime precisely correlated with the trajectory determined by our initial analysis (Figure 2—figure supplement 1)." "Our initial analysis" made me think again. They could clarify this is the sub-cluster analysis on the germ cell population.

Page7, line24-28, Seems to be a grammatical error/typo in this sentence.

Results, Page9, line4-5 and Figure 2H,I

"The second consisted of cells within larger clusters ({less than or equal to}8 cells), expressed relatively less foxl2l but also expressed rec8a" Some cysts that express foxl2l and rec8a have > 8 cells

What about stage 1a follicle cells, which surround the oogonial syncitial cyst, which cells are these or were they identified? Please comment in the manuscript.

The follicle cell clusters 3.1 and 3.2 identified are quite interesting. What stage are the 'early-stage germ cells'? Can you give a size?

Results, Page10, line24-28, and page14, line20-23

Reference styles are not consistent and complicated. In the first paragraph, they add the reference after each gene name, which is a much preferred style.

Figure 4, please note in legend what the arrows point to in the panels. The green nanos2 expression is difficult to discern from the blue in panel G, please point out.

Referring to stages as dpf, after 5 dpf, largely depends on feeding/growth conditions. The standard length (SL) of the fish is a much better way to report the stage for zebrafish >5 dpf, because it more accurately reflects the developmental stage than dpf. Stages can be quite different > 5dpf depending on feeding conditions. Could the authors also report SL of the fish used at 40 and 50 dpf, so that others can reproduce accurately the stage?

On page 20, paragraph 2, "Unfortunately, we have yet to identify where these cells reside in the ovary." Unclear why "unfortunately" is stated. Have the authors tested for the expression of particular genes in sub-cluster 3 of the stromal cells and not found it or just not tested? If the latter, then omit 'unfortunately'. If the former, then please provide additional information, as to what has been tested, as it could be helpful to readers.

On what basis do the authors ascribe HCR to being single molecule FISH. A reference should be given for the method. A reference should also be given or data shown that demonstrates the ability of the method to identify single molecules.

Figure 7A figure legend, "ligand-encoding genes" needs a more complete sentence here.

Results, Page22, line26-28, cyp11a2 should be cyp11a1.

---

## [Author Response]

Reviewer #1 (Recommendations for the authors):Some comments on presentation, typos, etc.

We thank the reviewer for their suggestions and for taking the time to identify typos.

P2l6 Would be helpful to the reader to have the new genes that are KOd listed in the abstract and will make the paper easier to find in an online search.

Excellent suggestion These have been added.

P4l24 add ref.

Reference added.

P10l1 something is missing here.

the “;” was a typo

P12l6 Medaka's work on germ cell stem cells should be referred to here. DOI:10.1126/science.1185473.

We are unclear why the referee is suggesting that this paper be referenced in this particular section, which deals with germ cell stage-specific gene modules, as the cited paper does not explore gene modules. To be certain, the Nakamura et al., (2010) paper is cited elsewhere, so we are not ignoring the contributions of this important work.

P13l1 should be 'First, nanos3 (FORMERLY called nanos1) is a …'

Corrected!

P23l22 These papers say that the receptor is cxcr4b, not cxcr4a. Erez Raz and Michal Reichman-Fried Attraction rules: germ cell migration in zebrafish. Bijan Boldajipour Cxcl12 evolution – subfunctionalization of a ligand through altered interaction with the chemokine receptor.

Thank you for catching this error. You are correct that *cxcr4b*=*odysseus*, not *cxcr4a*. This has been corrected.

Figure 7B, C. on the panel, indicate that magenta is stm, to match the other figures.

We have added this to Figure 7B as suggested

Figure 8. Text and legend need to say what cells formed the basis of the cells selected for subclustering.

This has been added as suggested. “To explore this further, we first extracted and re-clustered the cells identified in our preliminary cluster analysis as theca cells based on their expression of *cyp11a2* (Figure 1 and Figure 8).”

P36l20 check again cxcr4a vs b.

all have been corrected.

P35l18,19 The paper has several subclusterings, so here and elsewhere indicate which subclustering is referred to.

We now refer to the process re-clustering cells extracted from the initial cluster analysis as “re-cluster analysis” to distinguish it from sub-clusters, which are cells that likely represent cell subtypes,

P35l20 change 'per-follicle' to 'pre-follicle'.P38l14 "…mechanisms…are…"P38l27 Wtn4a◊Wnt4aP38l30 '…in other teleostS.'

the errors referenced above were in sections of the Discussion which have been deleted during the reorganization to combine the Results and Discussion.

P42 l21 check: '40 pairs of 40 dpf ovaries were dissected from Tg(piwil1:egfp) transgenic fish (Leu and Draper, 2010) were dissected and stored'.

This has been corrected.

Fig6 sup2 'subclsuter'.

This has been corrected.

Reviewer #2 (Recommendations for the authors):The manuscript is very well written and comprehensive, almost encyclopedic. However, the Results section is much too interpretative. This makes the Discussion section repetitive, as it mainly recapitulates points already made in the results. The lengthy results would be more concise if some of those points were addressed only in the discussion and this would greatly improve the legibility of the manuscript. Furthermore, it would be interesting to make parallels with other scRNAseq datasets from other organisms in the discussion. While some of the cell types like the oocyte progenitors may be unique to zebrafish, others, including stromal cell and theca cell subtypes closely parallel subtypes found in mammals which reinforces the value of the zebrafish as a model for the ovary.

We agree that making parallels with subtypes found in mammals reinforces the value of the zebrafish as a model and we in fact made a special effort to make these connections throughout the manuscript (e.g. all of the known stromal cell subtypes have clear mammalian counterparts, except the ovarian cavity epithelium (OEC). In the latter case, we note a similarity of the OCE gene expression with that of mammalian Mullerian ducts). In addition, and importantly, all of the markers we used to identify our cell populations were identified as markers of these cell types prior to scRNA-seq technology.

P8 line 10: There is a relatively small increase in PCNA expression in the putative oocyte progenitor cells. Do other markers of mitosis display the same pattern?

Yes. This is also true for some mitosis-expressed genes (e.g. lig1, mcm4, plk1, mki67). However, for others there is no difference in expression (e.g. dhfr). In this study we focused our analysis on identifying cell-stage specific markers, such as foxl2l. A more comprehensive analysis of cell cycle genes was outside the scope of our study.

Figure 2 I should read Nanos2, the gene name is clipped in the figure.

This has been corrected.

P20 line 20: What was attempted to identify the position of stromal progenitor in the ovary. What is the relevance of CD34 expression in these cells?

We have attempted to identify these cells using *pcolcea* as a probe. This gene was chosen because it is specific to this cell population and appeared to have relatively high expression. However, this probe has so far failed to work. We are currently producing alternative probes. The expression of *cd34* is very specific to the stromal progenitors in the ovary but the relevance of this is not known. However, consistent with our hypothesis, CD34 is known to be expressed in diverse progenitor cell populations in mammals (e.g. https://pubmed.ncbi.nlm.nih.gov/24497003/).

P21 line21. Mullerian duct epithelium and ovarian surface epithelium share a common embryonic origin in mammals. Were there further transcriptional similarities between OCE and mammalian OSE markers. What was the pattern of expression of AMHR2 in the zebrafish, which in mammals is expressed in both OSE and granulosa cells?

Good question. Zebrafish do not have an ortholog to Amhr. John Postlethwaite’s lab recently showed that at some point in the cypriniform lineage, to which zebrafish belong, there was a chromosomal translocation that resulted in the deletion of the ancestral *amhr* gene (Yan et al., 2019). Instead, there is very good evidence that the BMP type 2 receptor encoded by *bmpr2a* functions as the Amh receptor in zebrafish, as these mutants have a nearly identical phenotype to *amh* mutants (Zhang et al., 2020). Our data set shows high levels of expression of Bmpr2a in follicle cells, but little to no expression in other ovarian cell types, including the OSE.

P23 line15. It is peculiar that only a small subset of theca cells express StAR which would be required for the initial steps of steroid biosynthesis. Are they associated with more mature follicles?

We agree that this was unexpected. We have not localized the StAR expressing cells so we do not know if they are differentially localized to particular stage follicles. It is possible that the non-StAR expressing theca cells are immature while those that express StAR are the mature theca cells. However, while *cyp19a1a* mutant zebrafish are all males as adults, showing that E2 production is required for female development (Dranow et al., 2017), *star* mutant zebrafish have normal sex ratios, but females are sterile due to defects in oocyte maturation (Shang et al., 2019). These results argue that StAR is not essential for production of the E2 precursor androstenedione by theca cells in zebrafish. A discussion of these previous results have been added to the manuscript (Page 24, Lines 17-26).

The primary data, the code for its analysis, and the reagents employed were all clearly described and available as required by eLife's policies.Reviewer #3 (Recommendations for the authors):Text page 13, line 30, "Remarkably, we found that early-stage germ cells were always contacting lhx9+ pre-follicle cells". This is a bit confusing, as not all early-stage germ cells as seen in Figure 4 are in contact with the lhx9-expressing follicle cells. It seems that lhx9 follicle cells are located only on one side of a germ cell cluster. Please modify the text to make more clear.

Good point. We have modified this sentence as follows: “..we found that clusters of early-stage germ cells were always located adjacent to *lhx9*+ pre-follicle cells…” See also our response to the question regarding identification of the pre-follicle cell population below.

A number of the same points are made in the Discussion as in the Results section, including, for example, the conclusions made about E2 production. In the interest of shortening the paper and reducing redundancy, Results and Discussion could be combined into one section, with a final "Conclusion" section at the end for any additional points not discussed in the 'Results and Discussion" section. See also the point below.

We have modified the manuscript as suggested to reduce redundancy. The Results and Discussion are now a combined section.

Overall, this is a very long manuscript and might benefit by splitting it into 2 back to back papers to make it more digestible.

By combining the results and discussion, as suggested above, the length of the manuscript has been reduced to what we hope is an manageable size.

Abstract, Page 2, line3 "Our somatic cell data represents all known somatic cell types, including follicle cells, theca cells and interstitial stromal cells".It is a little confusing because the authors discuss the fibroblast-like interstitial stromal cells as one cell type of the stromal cells. In the abstract do the authors want to point out this subpopulation or might "ovarian stromal cells" be the more broad statement for the abstract here.

Thank you for this suggestion. We agree. A broader statement about stromal cells in the abstract is more appropriate. We have made this change to the abstract.

Introduction, Page 4, line23-24"E2 production also requires steroidogenic theca cells which produce androstenedione, the precursor that follicle cells use for E2 synthesis (ref)." Reference is missing.

The reference has been added.

Result, Page6, line9 (26), Tables. I couldn't find the Tables.

The Tables referenced are all available for download from DataDryad.org, as referenced in the Data Availability section. Regardless, we will also add them as supplemental data tables to make them more easily accessible.

Results, Page7, line21-22"We found that the directionality and the sequential gene expression along pseudotime precisely correlated with the trajectory determined by our initial analysis (Figure 2—figure supplement 1)." "Our initial analysis" made me think again. They could clarify this is the sub-cluster analysis on the germ cell population.

Good point. We have changed this sentence to say: “…correlated with the trajectory determined by our initial germ cell re-cluster analysis…”

Page7, line24-28, Seems to be a grammatical error/typo in this sentence.

We have modified this sentence to make it more understandable. “Notably missing from the stagespecific genes listed above are genes expressed in the proliferating oocyte progenitor cells, a population that is intermediate between the GSC and germ cells that have entered meiosis (sub-cluster 2 in Figure 2A).”

Results, Page9, line4-5 and Figure 2H,I"The second consisted of cells within larger clusters ({less than or equal to}8 cells), expressed relatively less foxl2l but also expressed rec8a" Some cysts that express foxl2l and rec8a have > 8 cells

Good catch! Thank you. That has been corrected to “≥4” cells.

What about stage 1a follicle cells, which surround the oogonial syncitial cyst, which cells are these or were they identified? Please comment in the manuscript.

Excellent point. This is addressed in answer to the following question.

The follicle cell clusters 3.1 and 3.2 identified are quite interesting. What stage are the 'early-stage germ cells'? Can you give a size?

Re-analysis of our *lhx9* in situ’s reveled that we can identify *lhx9+* cells surrounding clusters of *dmc1*+ Stage1A oocytes. Interestingly, and consistent with the scRNA-seq data, we now note that HCR in situs appear to identify a high *lhx9*-expressing cell populations, which are the cells that form the chord-like structures, and a low *lhx9*-expressing cell population, which form the remaining cells that appear to enclose the stage1A clusters. We have added a section to discuss this (page 16, lines 7-15) and propose that the high expression cells correspond to the cluster 3.1 subpopulation, while the low expressing cells correspond to the 3.2 subpopulation. We have modified Figure 4F to now include an example of this (see new Figure 4F’ and F”).

As to the point about the definition of “early germ cells,” we have better defined what germ cell-stage the various marker genes identify when these markers are first introduced in the germ cell section (page 7, lines 12-15). For example, we now state that *dmc1* is expressed in Stage 1A pre-follicle phase germ cells and that *zp3b* is expressed in Stage1B follicle phase oocytes. We also explicitly define “early-stage germ cells” in the *lhx9* section to mean “pre-meiotic and Stage 1A pre-follicle phase germ cells.”

Results, Page10, line24-28, and page14, line20-23Reference styles are not consistent and complicated. In the first paragraph, they add the reference after each gene name, which is a much preferred style.

We have modified as requested. In addition, we have added several other missing citations.

Figure 4, please note in legend what the arrows point to in the panels. The green nanos2 expression is difficult to discern from the blue in panel G, please point out.

We agree that this was hard to discern. We have added an inset in panel G that shows the *nanos2* and DNA channels only to highlight a *nanos2* expressing cell. We have modified the figure legend to reflect this addition. We also added to panel G two asterisks to identify auto-fluorescent red blood cells, and modified legend accordingly.

Referring to stages as dpf, after 5 dpf, largely depends on feeding/growth conditions. The standard length (SL) of the fish is a much better way to report the stage for zebrafish >5 dpf, because it more accurately reflects the developmental stage than dpf. Stages can be quite different > 5dpf depending on feeding conditions. Could the authors also report SL of the fish used at 40 and 50 dpf, so that others can reproduce accurately the stage?

We agree with the reviewer, but unfortunately, we did not keep track of fish length (SL). Instead, we have added to the M and M details about how we culture our fish (titled “Zebrafish Husbandry”).

On page 20, paragraph 2, "Unfortunately, we have yet to identify where these cells reside in the ovary." Unclear why "unfortunately" is stated. Have the authors tested for the expression of particular genes in sub-cluster 3 of the stromal cells and not found it or just not tested? If the latter, then omit 'unfortunately'. If the former, then please provide additional information, as to what has been tested, as it could be helpful to readers.

We attempted to identify these cells using an HCR probe set specific to *pcolcea*, since expression of this gene appears to 1) be very specific and 2) expressed at high levels in this sub-population. However, after several attempts we have not observed convincing signal. We do not favor identifying the genes we have tested but that did not work in our hands since we do not want to suggest that these genes are poor targets for probe design. It is possible that other probes targeting a different region of *pcolcea* would work. We have re-worded this sentence to say, “As we have yet to identify where these cells reside in the ovary, we do not know if these cells are a stable population that persist into adulthood or if they are a transient population present only during the juvenile and early adult stages while the ovary is increasing in size.”

On what basis do the authors ascribe HCR to being single molecule FISH. A reference should be given for the method. A reference should also be given or data shown that demonstrates the ability of the method to identify single molecules.

Excellent point. We should not have claimed that our method detected single molecules of RNA. All references to smFISH have been changed to “HRC RNA-FISH.”

Figure 7A figure legend, "ligand-encoding genes" needs a more complete sentence here.

This phrase has been deleted.

Results, Page22, line26-28, cyp11a2 should be cyp11a1.

Thank you for catching that! This has been corrected.